# Direct RNA sequencing of astronaut blood reveals spaceflight-associated m6A increases and hematopoietic transcriptional responses

Kirill Grigorev [1,2,10], Theodore M. Nelson [3,10], Eliah G. Overbey [1,2,4,5], Nadia Houerbi[1,2], JangKeun Kim [1,2], Deena Najjar [1,2], Namita Damle[1], Evan E. Afshin[1,2], Krista A. Ryon[1], Jean Thierry-Mieg [6], Danielle Thierry-Mieg[6], Ari M. Melnick [7], Jaime Mateus[8] & Christopher E. Mason [1,2,9] ✉

The advent of civilian spaceflight challenges scientists to precisely describe the effects of spaceflight on human physiology, particularly at the molecular and cellular level. Newer, nanopore-based sequencing technologies can quantitatively map changes in chemical structure and expression at single molecule resolution across entire isoforms. We perform long-read, direct RNA nanopore sequencing, as well as Ultima high-coverage RNA-sequencing, of whole blood sampled longitudinally from four SpaceX Inspiration4 astronauts at seven timepoints, spanning pre-flight, day of return, and post-flight recovery. We report key genetic pathways, including changes in erythrocyte regulation, stress induction, and immune changes affected by spaceflight. We also present the first m⁶A methylation profiles for a human space mission, suggesting a significant spike in m⁶A levels immediately post-flight. These data and results represent the first longitudinal long-read RNA profiles and RNA modification maps for each gene for astronauts, improving our understanding of the human transcriptome's dynamic response to spaceflight.

Both short- and long-term missions into space are correlated with a variety of acute physiological effects, affecting diverse organ and cellular systems[1–3]. Astronauts generally experience performance impairment in a wide variety of activities post-spaceflight, requiring reconditioning routines to regain preflight performance[4]. Significant changes also occur within the circulatory and hematological system during spaceflight, on both a molecular and physiological level, which likely contributes to these effects[5]. Ground studies collecting transcriptomic profiles of human subjects have attempted to characterize the impact of individual components of spaceflight, such as radiation or microgravity exposure[6,7], the latter being simulated with a parabolic

flight[8]. With regard to direct spaceflight exposure, the available transcriptomic data is even more limited, focused on a few specific assays, either sequencing the human IgM repertoire following long-term spaceflight[9], measuring the proteome of exhaled breath condensate after long-term spaceflight[10], or examining gene expression profiles from astronaut hair follicle samples after long stays on the International Space Station[11]. However, beyond this limited work, there is scant data on the impact of human spaceflight on the transcriptome beyond the NASA Twins Study (1-year mission), and almost no data from short-duration missions. Moreover, there are no single-molecule, direct RNA datasets from astronauts, which can help detail changes in

¹Department of Physiology and Biophysics, Weill Cornell Medicine, New York, NY, USA. ²The HRH Prince Alwaleed Bin Talal Bin Abdulaziz Alsaud Institute for Computational Biomedicine, Weill Cornell Medicine, New York, NY, USA. ³Department of Microbiology and Immunology, Vagelos College of Physicians and Surgeons, Columbia University Irving Medical Center, New York, NY 10032, USA. ⁴Center for STEM, University of Austin, Austin, TX, USA. ⁵BioAstra, Inc, New York, NY, USA. ⁶National Center for Biotechnology Information (NCBI), National Library of Medicine, NIH, Bethesda, MD 20894, USA. ⁷Department of Medicine, Weill Cornell Medicine, New York, NY 10021, USA. ⁸Space Exploration Technologies Corporation (SpaceX), Hawthorne, CA, USA. ⁹WorldQuant Initiative for Quantitative Prediction, New York, NY, USA. ¹⁰These authors contributed equally: Kirill Grigorev, Theodore M. Nelson. ✉e-mail: chm2042@med.cornell.edu

RNA methylation and base modifications while also capturing expression and isoform changes.

To address this gap in transcriptome knowledge, eleven different biospecimen types underwent four types of high-throughput sequencing for each of the four civilian astronauts on the Inspiration4 (I4) mission[12], including genomics, transcriptomics, immune profiles, microbiome, and some clinical blood tests[13]. Long read sequencing (LRS) technologies have potentially relevant clinical applications[14], since they can capture both splicing and expression differences in full-length transcripts; nanopore long-read direct-RNA sequencing[15] is additionally advantageous since it provides per-base epitranscriptomic data (e.g., $m^6A$ modification sites)[16]. Here, we generated LRS samples taken before the flight (L-92, L-44, L-3 days), the day of landing (R + 1), and in recovery periods (R + 45, R + 82, R + 194 days) after a 3-day spaceflight from the SpaceX Inspiration4 mission. A battery of computational and analytical approaches were deployed on these samples, including edgeR, DESeq2, ONT tools, Gene set co-regulation analysis (GESECA)[17–19], as well as orthogonal sequencing on the Ultima UG100 platform, which generated an average of 413 million reads per sample. These data provided an opportunity to discern the utility of LRS technologies in studying the I4, as well as future astronaut cohorts, and will also enable meta-analyses in conjunction with other NASA Open Science Data Repository (NASA OSDR, https://osdr.nasa.gov/) short-read and single-cell data sets.

We present these data sets and algorithmic methods as foundational functional genomics resources for human spaceflight, as part of a roadmap for future bioinformatics data analysis techniques for LRS data. These current results are in line with the mission of NASA OSDR, which aims to standardize procedures for future spaceflight missions and open science, enabling comprehensive comparisons between these astronaut and model cohorts[20]. We also implemented methodologies which allow for examination of differences in expression, splicing behavior, and methylation ($m^6A$) at the same time. As such, we can observe molecular changes underlying blood transcriptome dynamics across an astronaut's journey to space and back, which show thousands of changes across several hundred specific genes, some of which recapitulate known stress markers of the vasculature and immune system, as well as signatures of $m^6A$ changes, isoform switching, and hematopoiesis regulation that appear upon landing back on Earth.

## Results

### Differential gene expression and gene set co-regulation analyses reveal genes and pathways associated with the effects of short-term space flight

We evaluated two distinct approaches to gene expression quantification and detection of differentially expressed genes (DEGs) from the direct RNA data (see "Methods"). First, the raw basecalled reads were processed with the pipeline maintained by Oxford Nanopore Technologies (ONT), pipeline-transcriptome-de[21]; the pipeline considers reads aligned to the Gencode v41 transcriptomic reference[22], includes multi-mapping reads, quantifies expression based on individual transcripts, collapses transcript counts into gene counts, and obtains sets of DEGs using the edgeR package[17]. In addition, we quantified expression based on reads aligned to the human genome reference hg38[23] using the subread package[24], excluding multi-mapping reads, and obtained sets of DEGs via the SARTools package[25], a wrapper for DESeq2[18]. Both pipelines were applied to the data in the context of multiple flight, recovery, and longitudinal profiles (Supplementary Data 1). A full flow cell was used for each astronaut and time-point; the four astronauts were grouped together as biological replicates for downstream analysis. All runs aligned more than 400,000 long-reads, with an average length of 580 base pairs. While the two computational approaches produced results that slightly differed in DEGs determined to be significant (DESeq2 or edgeR $p$ value < 0.05) (Supplementary

Data 2), they overwhelmingly agreed on their relative abundance estimates for gene quantification (median Pearson's $r = 0.98$, Fig. 1a) and the direction of their change in all profiles (Fig. 1b).

Next, gene set co-regulation analysis (GESECA)[19] was used to examine changes in known pathways and molecular functions (MSigDb category C2, curated gene sets)[26,27], specifically for those pathways deviating from the baseline value (median z-scored expression of associated genes +1.86, Supplementary Data 3) at the R + 1 timepoint (immediately post-flight). The variation of 789 out of the 6336 assessed pathways was significantly co-regulated (adjusted $p$ value < 0.05). Although 2829 downregulated genes were present at R + 1 (z-score below 0), the vast majority of significantly variable pathways was driven by upregulated genes (z-score above 0, Supplementary Fig. 1). Top variable pathways included erythrocyte $CO_2$ and $O_2$ takeup and release, as well as the Biocarta AHSP pathway (hemoglobin's chaperone; a superset of the erythrocyte-associated pathways) (Fig. 2). Longitudinal changes for the identified differentially enriched pathways showed two distinct trends (Supplementary Data 3, value of $dz$); the trend was either disrupted immediately following space flight (Fig. 3a) or was continued from pre-flight before returning to baseline in recovery (Fig. 3b). While most altered heme pathways were marked by increased expression of genes when returning to Earth, two pathways for erythrocytes showed distinct responses. Specifically, the data demonstrated both a decrease in erythrocyte-associated pathways for erythrocyte $CO_2$ and $O_2$ takeup (and release), as well as an increase in Steiner membrane genes, which indicates the significant spaceflight-induced stress on erythrocytes in particular.

Transcription factor enrichment analysis was then utilized to identify commonly shared transcription factors by genes (Supplementary Data 4) across all time points. The top three suggested regulators, based on a combined CHIP-seq and co-expression analysis, *KLF1*, *GATA1*, and *TAL1* are all key transcription factors (TFs) for erythrocyte differentiation[28–30], underscoring the changes in these cell types and also suggesting possible TF-based drivers for the trends observed within the GESECA analysis.

### $m^6A$ modification analysis identifies sites of differential longitudinal methylation

Next, we identified the sites of $m^6A$ modification across isoforms[31] and their degree of change, leveraging the single molecular nature of the ONT platform to find sites differentially methylated across the pre-flight, recovery, and longitudinal post-flight profiles ($q$ value < 0.01, methylKit[32]) (Supplementary Data 5). This analysis showed 1190 $m^6A$ modification sites with statistically significant differences for at least one particular longitudinal comparison. The significant sites were annotated to 200 total transcripts, sourced from 193 gene loci. Notably, the largest set of differentially methylated positions occurred on the day of returning back to Earth (timepoint R + 1 compared to all pre-flight timepoints), which we correlated with the variability of pathways identified by GESECA. Per pathway, the number of differentially methylated sites per gene ranged from 0 to 4.4 (Fig. 4a). Among the pathways with the highest density of differentially methylated sites, once again the erythrocyte-associated pathways were enriched, as well as their superset, the Biocarta alpha-hemoglobin stabilizing protein (AHSP) pathway (Fig. 4b).

Comparing the 193 gene loci which contained DRACH (D = A, G or U; H = A, C or U) sequence motifs[33] with differentially methylated sites (minimum 20x coverage, $q$ value < 0.01, methylKit) with the set of 440 differentially expressed gene loci (FDR-adjusted $p$ value < 0.05, either salmon or SARTools), we found an overlap of 53 genes, indicating that most of the post-transcriptional $m^6A$ regulation is distinct from the set of DEGs. When individual genes were considered, the genes with the highest number of differentially methylated sites did not contribute to significantly differentially expressed pathways, but among the top 30 such genes, 11 were part of the HSIAO_HOUSEKEEPING_GENES pathway

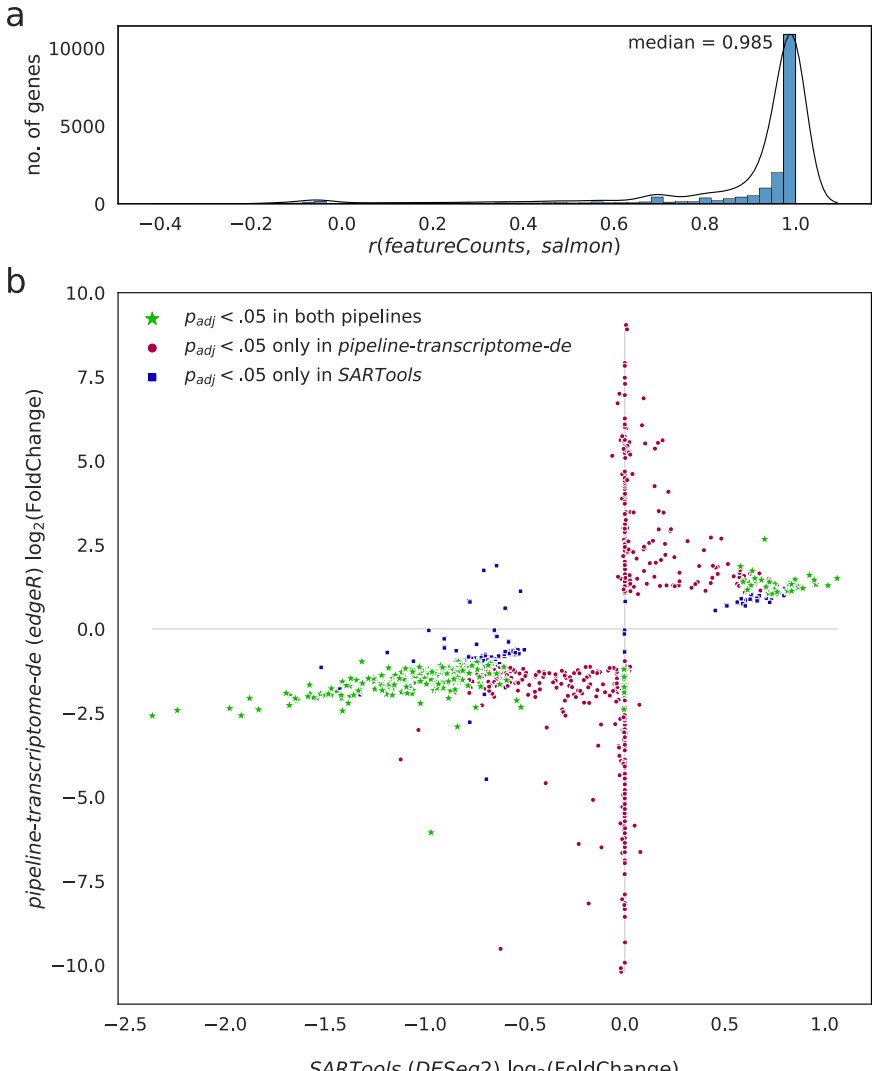

**Fig. 1 | Comparison of DGE pipelines.** Correlation of (**a**) per-gene counts obtained from featureCounts and salmon, and of (**b**) $\log_2$(FoldChange) values obtained from SARTools and pipeline-transcriptome-de. $p$ values are obtained from a Wald test (DESeq2) and a generalized linear model test (edgeR), respectively, and plotted following FDR adjustment; in green, significant ($p$ value < 0.05) as reported by both pipelines, in red, only reported as significant by pipeline-transcriptome-de, and in blue, only reported as significant by SARTools.

(Supplementary Fig. 2), and included genes for alpha-globin (*HBA*), beta-globin (*HBB*), and also human leukocyte antigen (*HLA*) alleles. These data indicate the transcription levels of the genes responding to spaceflight are distinct from their specific RNA modifications, and that this is consistent across the immune and hematopoietic-related pathways.

We next examined the m⁶A differences observed within the flight profile (FP) and return profile (RP) comparisons (FP1, RP1, and RP2, defined in Supplementary Data 1). Overall, 331 positions were hypermethylated ($q$ value < 0.01, methylKit) and, among these, seven genes (*PFN1*, *LAPTM5*, *RPS4X*, *HBA1*, *MT-ND4*, *HBA2*, *HBA2.1*) demonstrated a statistically significant downregulation in methylation 45 days after landing ($q$ value < 0.01, methylKit). An additional two transcripts (*MT-ND1*, *MT-RNR1*) demonstrated a statistically significant downregulation 82 days after landing ($q$ value < 0.01, methylKit). 208 positions were noted to contain decreased levels of m⁶A ($q$ value < 0.01, methylKit). Amongst these, one (*SLC25A37*) position demonstrated increased methylation 45 days after landing ($q$ value < 0.01, methylKit). No additional positions demonstrated increased methylation ($q$ value < 0.01, methylKit) in the long-term follow-up timepoints, which indicates that spaceflight has its most

significant effect on m⁶A methylation levels in the days immediately following return to Earth.

## De novo transcriptome analysis reveals uniquely expressed transcripts at L-92, L-3, R + 1 and R + 194

De novo transcriptome analysis was then utilized to reveal transcripts unique to particular time points. Unlike differential expression analysis, which captures differences across timepoints based on known annotations, de novo transcriptome analysis captures unannotated transcript variants specific to a particular timepoint or sample type. Ignoring loci not detected within the samples, we note that the overall sensitivity of the full transcript-level analysis, as defined by GffCompare[34], is on average 8.5%, revealing 3′ enrichment of many reads (Supplementary Data 6), which is expected given the polyA-priming protocol. One particular sample (November for the C003 astronaut) had lower coverage, leading to the exclusion of this time point from subsequent analysis. Additionally, when ignoring completely unannotated putative transcripts, we report that the average precision, as defined by GffCompare, was 52.4% (Supplementary Data 6). We focused on transcripts found consistently across all crew members for subsequent de novo analyses.

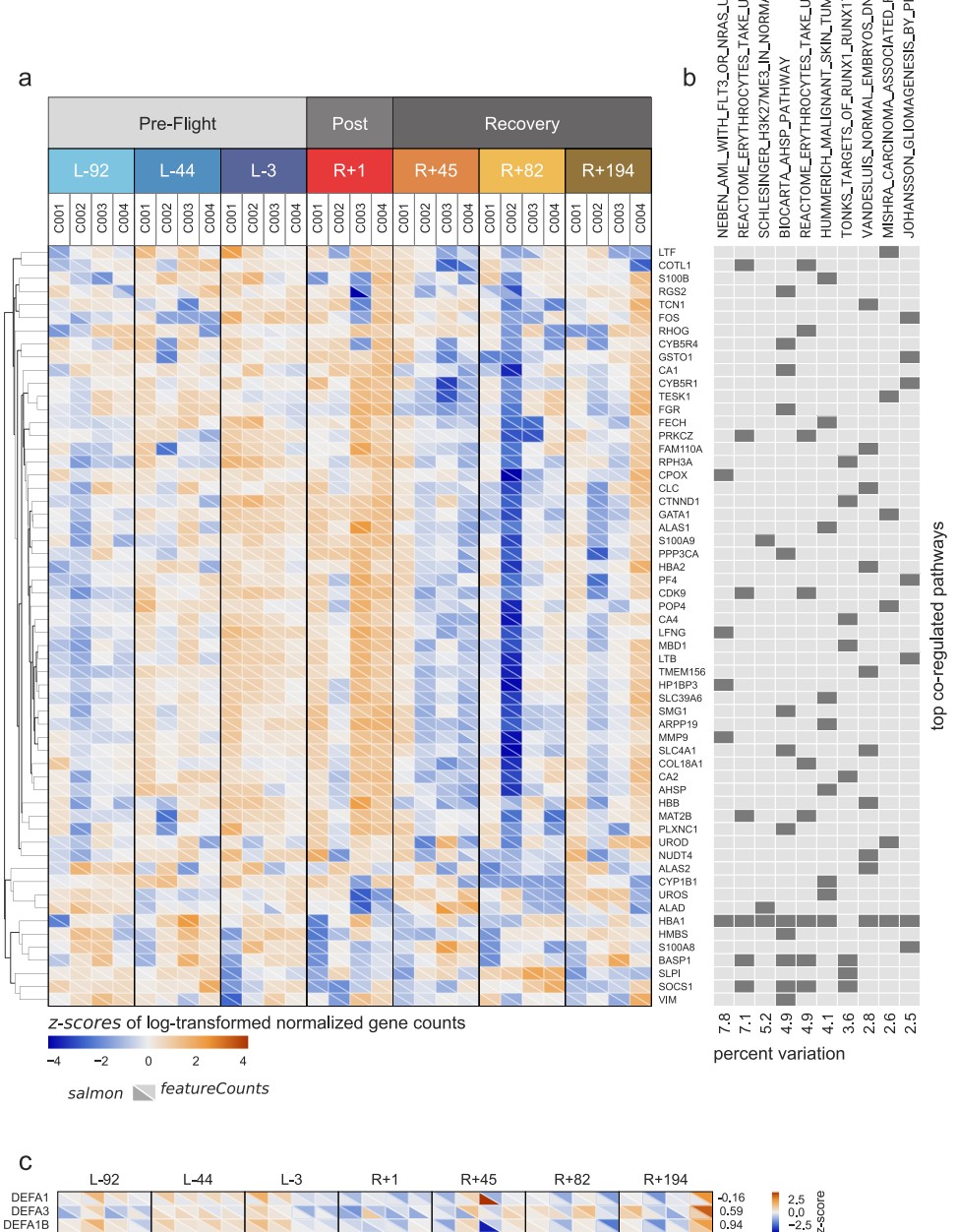

**Fig. 2 | Patterns of gene expression in the top variable pathways. a** z-scored expression values of genes from the top 10 pathways most variable across all timepoints, as reported by featureCounts and salmon. **b** The top 10 pathways; dark rectangles denote the genes belonging to each pathway. Gene sets can be explored further at: https://www.gsea-msigdb.org/gsea/msigdb. **c** An example of the difference in the attribution of read counts to homologous genes when discarding secondary mappings (featureCounts) and when partially accounting for them (salmon); Pearson's r between the counts reported by the two tools is annotated to the right of the heatmap.

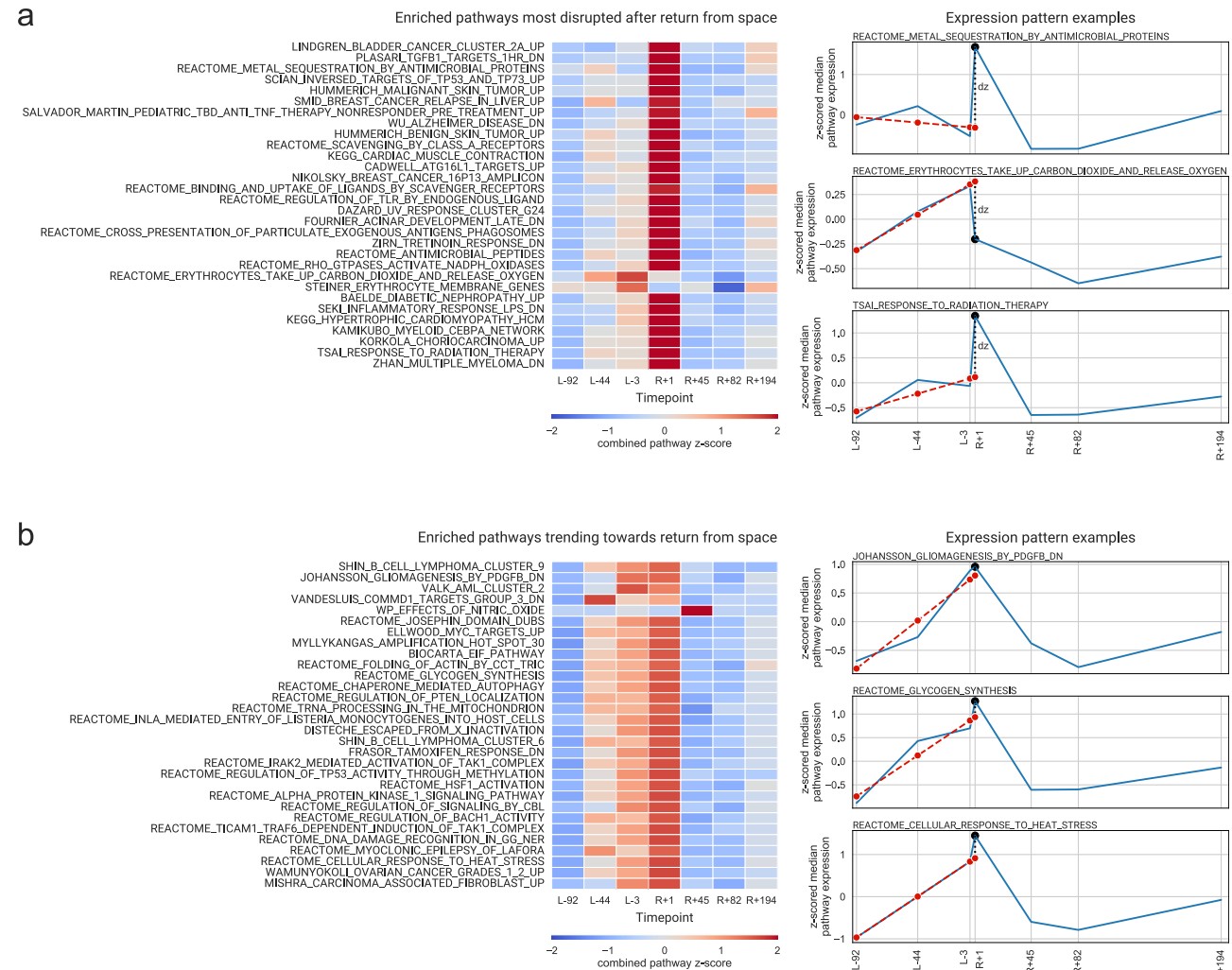

**Fig. 3 | Patterns of pathway co-regulation in the course of pre-flight, return, and recovery. a** Top 30 pathways most disrupted after return from space. Gene sets can be explored further at: https://www.gsea-msigdb.org/gsea/msigdb. **b** Top 30 pathways most consistent in expression during both pre-flight and return from space. On the right: examples of calculation of *dz*, the deviation of the *z*-score from the value expected under the assumption that the pathway is not disrupted.

The transcripts were then analyzed in the context of two overlapping time periods: (1) the period leading up to the space flight and return; and (2) the period starting with spaceflight and leading into post-flight recovery. Therefore, first, the set of samples consisting of preflight and return timepoints (L-92, L-44, L-3, and R + 1) was examined. There were eight unique transcripts for L-92 (Supplementary Data 8: June), no unique transcripts for L-44 (Supplementary Data 8: August), seven unique transcripts for L-3 (Supplementary Data 8: September Pre-Flight) and six unique transcripts for R + 1 (Supplementary Data 8: September Post-Flight, top). In the set of samples consisting of return and recovery timepoints (R + 1, R + 45, R + 82, and R + 194), there were five unique transcripts for R + 1 (Supplementary Data 8: September Post-Flight, bottom), one unique transcript for R + 82 (Supplementary Data 8: December) and no unique transcripts for R + 194 (Supplementary Data 8: March). These unique RNAs represent perhaps a very specific response to spaceflight, which have not been observed in previous studies.

Since spaceflight can introduce changes in differential expression and methylation, we next examined if the stress could also lead to aberrant splicing. Each transcript was assigned a GffCompare transcript classification code, which represents the detection of intron retention or other changes in isoform state[34]. The i4 transcriptome data showed six transcripts with complete intron chain retention ("=" label), nine transcripts with intron chain retention over a segment ("c"

label), one transcript with single intron retention ("n" label), five transcripts characterized by a unannotated intron ("j" label), one transcript with a single exon fragment partially covering an intronic region ("e" label), two transcripts with unannotated exons ("o" label), and three transcripts entirely unannotated and not found in any reference transcriptome ("u" label). These results (Supplementary Data 8) show the capacity for novel isoform state and discovery in the direct RNA-seq data.

Of note, we also identified two transcripts which could be uniquely associated with the R + 1 timepoint (Supplementary Data 8), and downregulated during the recovery phase upon returning to Earth. The first of these was *ENST00000686344.2*, within the *ENSG00000274015* gene locus, located on chr14 between coordinates 63,642,035 and 63,665,593, which is an uncharacterized lncRNA annotation. The second is a completely novel, unannotated putative transcript variant, described by the "u" *GffCompare* transcript classification code[34], located on chromosome 10 between coordinates 3,408,655 and 3,409,052 (Supplementary Fig. 3).

To confirm these isoforms, we performed deep RNA-seq on the same whole blood samples using the Ultima Genomics UG100 system. The platform generated an average of 411.5 million reads per sample, with a mapping percentage greater than 90% for all samples (Supplementary Data 6). As a result, an average of 28,438 genes were detected, with close correspondence between all samples (Fig. 5A). When

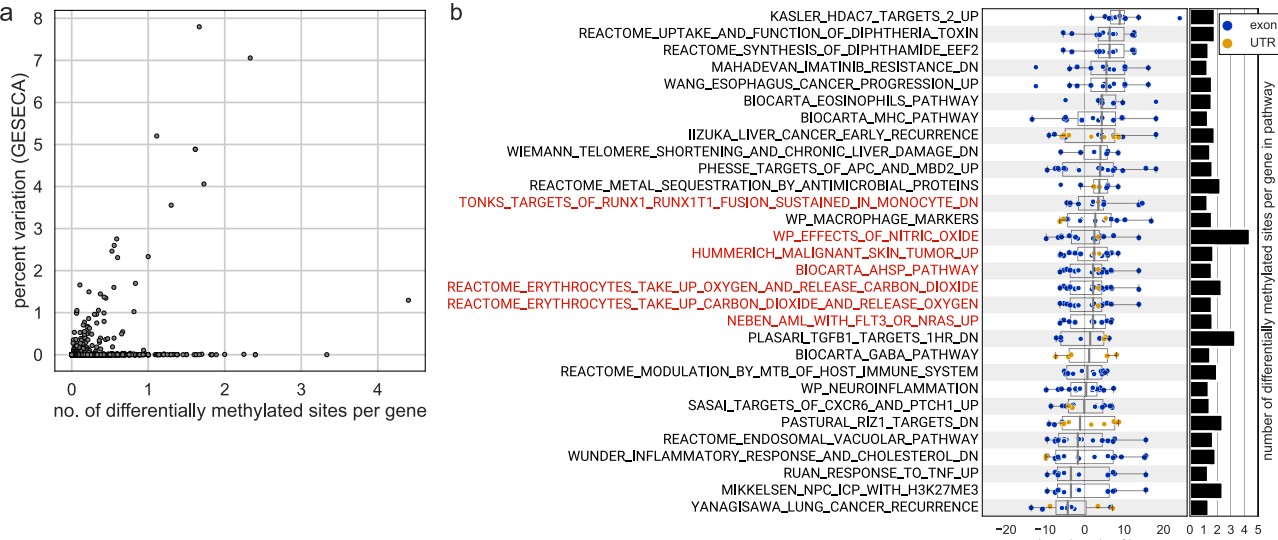

**Fig. 4 | Pathways with most differentially methylated genes. a** Correlation of the number of differentially methylated sites per gene, and expressional variability of each pathway. **b** Top 30 pathways with the most differentially methylated sites per gene, ranked by median differential methylation. Each dot in the box plot represents the value of differential methylation percentage of a site on any gene in the given pathway; the site's location in the exon or the UTR is color-coded. Boxes represent the range between Q1 and Q3 (the first and the third quartile) of the distribution, thick lines within the boxes annotate the median value, and the whiskers extend by 1.5×(Q3 − Q1) in each direction. Pathways that are also significantly variable by expression (GESECA adjusted $p$ value < 0.05; obtained from a permutation test) are annotated in red. Gene sets can be explored further at: https://www.gsea-msigdb.org/gsea/msigdb.

comparing the nanopore quantification of gene expression profiles to the Ultima data, we observed fair correlation ($R^2$ > 0.56−0.71) with the UG100 platform (Fig. 5B). We determined that 23/25 (92%) of the time point specific genomic loci demonstrated a consistent pattern of cDNA read alignments (Supplementary Data 6). While this did not include the transcript variant on chromosome 10 within Supplementary Fig. 3, we were able to confirm the expression of the uncharacterized R + 1 associated lncRNA transcript *ENST00000686344.2*, within *ENSG00000274015* gene locus (Fig. 5C).

## Discussion

All prior transcriptomics data from spaceflight studies have used short-read sequencing[14], and these LRS data showed a unique signature of disrupted gene regulation for astronauts from the Inspiration4 mission, particularly in the hematological system. These data provide the first map of m6A changes from spaceflight, which showed their largest degree of changes on the day of landing (R + 1)[15,16]. The majority of significantly altered pathways was driven by upregulated genes, implying that transcriptional and pathway activity went into figurative "overdrive" following the short-term, high-elevation space flight. This significant shift in the expression profile may be explained by both direct and indirect effects of spaceflight, such as radiation, changes in gravity, circadian rhythm disruptions, mission stress, and possibly medications, although none were indicated by the crew records. The direct-RNA sequencing data also detailed splicing events for isoforms, while confirming that aberrant splicing in the crew was very low overall.

Although many categories of significantly up-regulated pathways were observed, exposure to space radiation in particular may underlie a number of cancer-related pathways, including "TSAI response to radiation therapy" and "Wiemann telomere shortening and chronic liver damage." The set of downregulated pathways after landing was distinct, and notably included genes associated with breathing regulation (e.g., $CO_2$ and $O_2$ takeup and release by erythrocytes). These pathways were also among the top differentially methylated, suggesting a potential convergence between expression and methylation in regulating these critical functions. These results also mirror

reported effects of partial gravity and microgravity on lung function, and spaceflight-related hypercapnia linked to increased $CO_2$ levels[35,36]. Future studies should consider the impact of air quality, convection, and microgravity-related pulmonary challenges for expression and methylation differences. Also, while these samples had to be sequenced on Earth, the development of space-specific, microgravity-compatible sequencing protocols[37,38] foreshadows the exciting possibility of real-time monitoring of changes of the transcriptome and epitranscriptome for future missions. Moreover, radiation levels should continue to be monitored during missions, since such stressors can alter telomere dynamics, which have been observed before in the NASA Twins Study[39,40] and for mountaineers climbing Mt. Everest[25].

In addition to capturing differential expression, LRS technologies also capture transcript splicing, potentially allowing for the identification of new regulatory mechanisms and interactions between these two modalities of RNA[41]. In fact, a recent study examining murine skeletal muscle post-spaceflight found that differential splicing accounted for more variation than differential expression[42]. LRS can potentially identify all such differences, given that the nearly all transcripts are shorter than the theoretical maximum sequencing length for Nanopore platforms (2.3 Mb)[43,44]. Nevertheless, limitations remain within the library preparation and technical specifications (e.g., 3′ enrichment) of different LRS platforms[43], and improvements in these protocols and informatics methods will help profiling for future samples.

To gain greater confidence in the DEG set, we compared two different approaches to multi-mapping reads and DEG detection. Namely, secondary mappings are not included in the featureCounts quantification, whereas secondary mappings contribute partial counts in salmon, and are likely the source of differences in assigning the counts to three closely homologous genes *DEFA1/1B/3* (Fig. 2c). Additionally, the associated differential expression callers, DESeq2 and edgeR, respectively, utilize slightly different heuristics to perform $\log_2$ fold change estimates, resulting in many genes uniquely identified by the salmon-edgeR pipeline mapping to the *y*-axis of Fig. 1b. While accounting for secondary mappings with salmon could be argued to be the more sound approach, this calls for further assessment and

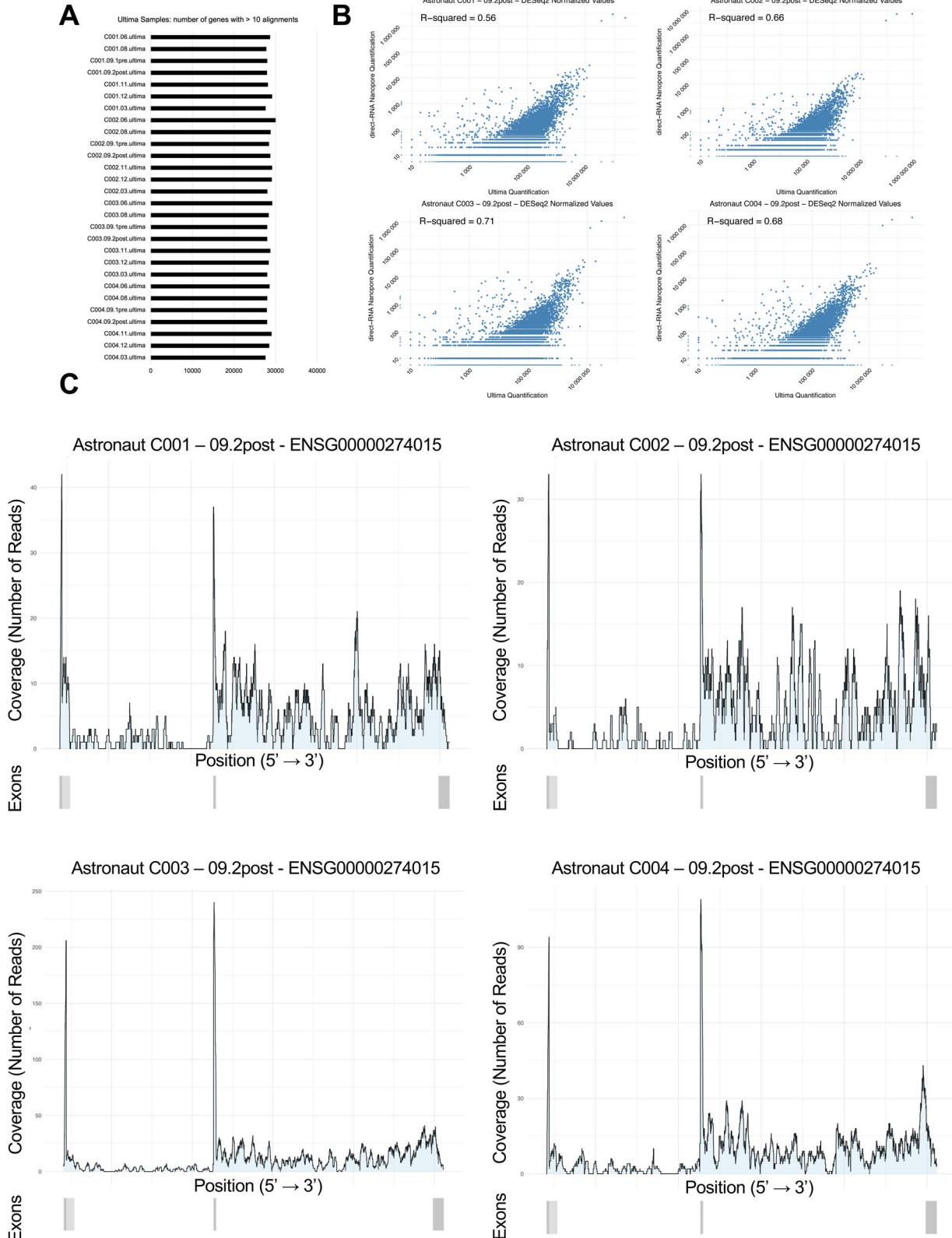

**Fig. 5 | Validation of transcript variants utilizing Ultima Genomics deep RNA-sequencing. A** Number of genes detected by each Ultima sample, defined to have more than 10 alignments within the genomic locus. **B** Correlation between normalized DESeq2 gene expression profiles of R + 1 samples produced by either Nanopore direct-RNA Promethion or Ultima Genomics UG100 platform. **C** Coverage visualization for *ENSG00000274015* for R + 1 samples. The gene structure is described by a single panel beneath the coverage plot, where darker shades of gray represent multiple merged transcript variants.

discussion of methods to be used for analyses of direct-RNA data, as retention of secondary mappings is incompatible with m6A callers such as m6anet, which was recently shown to be the most accurate m6A caller[45]. Despite these differences attributed to isoforms, we nonetheless have found that both methods produced similar quantification results, with high correlation (>0.98), and their per-gene averages were used for gene set co-regulation analyses (GESECA). When the significance of differential expression of individual genes is to be considered, intersecting the gene sets reported by DESeq2 and edgeR was preferred.

A current limitation specific to nanopore LRS is the loss of coverage at the 5' end of transcripts, since the assay uses polyA-trail priming[46,47]. As expected, we observed this truncation, resulting in lower sensitivity for some reference transcripts (Supplementary Data 6). It has also been suggested that future improvements in the Nanopore hardware will address this limitation[48]. To minimize this issue, those with other classification codes were restricted to those events observed across all crew members and all timepoints, and thus likely represent real biological variation. Moreover, we verified the associated transcripts with orthogonal data from the Ultima Genomics UG100 platform. While these samples provide remarkable coverage, further comparative analysis is necessary to optimize the application of current differential expression models. Also, transcript-specific and functional follow-up assays are necessary to determine the potential significance of these findings and potential use of these transcripts as biomarkers, as well as additional data from other crews and missions. Moreover, given the civilian background of the I4 astronauts, we note that transcriptomic changes can be confounded with age, biological sex, preflight preparation differences, or divergent postflight recovery procedures, and thus utilized tools that aim to model and normalize such variation.

Finally, given the recently proposed missions with private (SpaceX, Axiom, Sierra Space) and public (NASA, ESA, and JAXA) space entities, there will be continued opportunities to replicate these findings and to continue to discover new features of the transcriptome. This will include continued mapping of space-related gene expression responses, m6A site changes, and isoform-switching events that are associated with spaceflight. Also, in-flight testing of direct RNA sequencing has been demonstrated on the ISS, which indicates that LRS of RNA could also be applied for crew studies in future missions, including for blood, microbial, and environmental samples. Optimization and deployment of these single molecule technologies can also aid in the plans for lunar laboratories, in-flight clinical diagnostics, and the ability to discover non-canonical bases or new base modifications for lunar and exploration-class (e.g., Mars) missions.

## Methods

### IRB statement

The study design conforms with all relevant ethical regulations related to human subject research. All subjects were consented at an informed consent briefing (ICB) at the SpaceX headquarters (Hawthorne, CA), and samples were collected and processed under the approval of the Institutional Review Board (IRB) at Weill Cornell Medicine, under Protocol 21-05023569. All crew members have consented for data and sample sharing. The study was conducted in accordance with the criteria set by the Declaration of Helsinki. Participants were not compensated for their participation in this study.

### Direct RNA sequencing

Total RNA was processed and sequenced as described in Collection of Biospecimens from the Inspiration4 Mission Establishes the Standards for the Space Omics and Medical Atlas (SOMA)[12]. Briefly, total RNA was isolated using the direct-RNA kit (Oxford Nanopore Technologies) from whole blood samples from three pre-flight time

points L-92, L-44, L-3, one post-flight time point R + 1, and three recovery period time points R + 45, R + 82, and R + 194. Basecalling was performed with Guppy version 6.2.1, and alignment of raw nanopore events to the basecalled sequences was done with the f5c module eventalign version 1.1[49]. Quality assessment was performed with pycoQC version 2.5.0.21[50] and MultiQC version 1.13.dev0[51] (Supplementary Data 7).

### Differential gene expression analysis

The reads were aligned to the Gencode v41 human transcriptomic reference[22] with minimap2 version 2.24-r1122[52]. First, the pipeline maintained by Oxford Nanopore Technologies, pipeline-transcriptome-de[21], was used to quantify transcripts and to subsequently perform the differential expression analysis. pipeline-transcriptome-de performs the quantification with salmon[53], accounting for multi-mapping reads; therefore, we included the highest-scoring secondary minimap2 alignments (minimap2 switch -p1.0) in the input for the pipeline. Additionally, the sub-2Kbp read lengths (median length between 517 and 1331, with one outlier of 279, Supplementary Data 7) were insufficient to perform differential transcript usage analyses[47], and we focused on the differential gene expression results, which pipeline-transcriptome-de generates with edgeR[17] while collapsing transcripts into respective genes.

On the other hand, however, we also planned to perform differential methylation analyses, and tools that detect m6A modifications, such as m6anet[31], require that secondary alignments be filtered out. To (1) match the input requirements of m6anet that prohibit multi-mapping reads, (2) compare the collapsed edgeR results to a pipeline that performs gene quantification by design, and (3) assess the effect of aligning reads to the whole genome rather than to the transcriptome, we separately aligned the reads to the human reference genome hg38[23] with minimap2 with the parameters recommended for direct-RNA data, retaining only the primary mappings (-x splice -uf -k14 --secondary=no). We thereafter processed these data with the subread module featureCounts version 2.0.1[24] to perform gene quantification, and subsequently analyzed differential expression with SARTools version 1.8.1[25], a package wrapping DESeq2 version 1.36.0[18]. As DESeq2 and edgeR employ different approaches to $\log_2(\text{FoldChange})$ shrinkage, the determination of significance was based solely on the magnitude of the $p$ values reported by both; only the genes that were identified as significantly differentially expressed by both tools (FDR-adjusted $p$ value < 0.05) were considered as such. Covariates to account for inter-astronaut variability were included when performing both DESeq2 and edgeR analysis. For comparisons where multiple time-points were grouped as either preflight or postflight, each sequence run was treated as an independent replicate.

### Gene set co-regulation analysis

Given high correlation of results between featureCounts and salmon, per-gene counts were averaged between the two tools. These values, normalized by library size, were supplied to GESECA[19], and co-regulation of MSigDb C2 pathways (curated gene sets)[26,27] was inferred. Pathways were deemed significantly variable if the reported adjusted $p$ value fell below 0.05. For each significantly variable pathway and at each timepoint, the median $z$-scored expression value of genes in the pathway was calculated. Next, to determine the pattern in the pathway variability upon return from spaceflight, per pathway, we fitted a linear regression on the median $z$-scores at preflight timepoints as a time series (i.e., values of −92, −44, −3), obtained a prediction of the $z$-score at the return timepoint (+1) and calculated a value, $dz$, of how much the observed value at R + 1 differed from the predicted value; examples of this approach are illustrated on Fig. 3. The pathways were ranked by this value, from highest (most disrupted by spaceflight) to lowest (continuing a preflight trend).

## Transcription factor enrichment analysis

Gene annotations within Supplementary Data 2 were converted to gene names utilizing gProfiler (https://biit.cs.ut.ee/gprofiler/convert)[54]. The resulting gene list was input into the ChEA3 digital web server (https://maayanlab.cloud/chea3/)[55]. The mean rank results were exported and reported in Supplementary Table 4.

## m⁶A modification analysis

The transcriptomic alignments were filtered to retain primary mappings only (minimap2 switch --secondary=no) to match the input requirements of m6anet. m6anet version 1.1.0 was then employed to annotate sites of m⁶A modifications with methylation probabilities, and methylKit version 0.99.2[32] was used to discover differentially methylated sites between conditions in all profiles; sites were deemed significant if the resultant methylKit $q$ value fell below 0.01, as recommended by the tool's authors.

## De novo transcriptome analysis

The genomic alignments were collapsed with StringTie version 2.2.1[56] and compared with GffCompare version 0.11.2[34]. The GffCompare.-tracking file was further processed in R version 4.2.1/RStudio version 1.2.5001. Average read length was calculated with samtools version 1.16.1[57]. The R + 45 timepoint was excluded since the sample for subject C003 exhibited a low average read length (279 bp) and transcript-level precision (Supplementary Data 7).

## Ultima genomics deep-RNA sequencing

Poly-A mRNA was isolated from approximately 50–500 ng of total RNA using the NEBNext Poly(A) mRNA Magnetic Isolation Module (New England Biolabs). The resulting mRNA was converted into cDNA using the NEBNext Ultra II Directional RNA Library Prep Kit (New England Biolabs). After end repair/dA-tailing, adapter ligation, and USER-enzyme digestion, indexing PCR was carried out using the UG Library Amplification Kit (Ultima Genomics) with custom indexing primers that amplify the TruSeq Read1 and Read 2 sequences, but are appended with Ultima Genomics adapter sequences. After 11 cycles of PCR, amplification products were cleaned up using a 1.0x ratio of AMPure XP beads (Beckman Coulter). Overhang generation was carried out using components from the UG Library Amplification Kit (Ultima Genomics) and material was size selected using a dsSPRI approach with 0.6x/0.85x AMPure XP beads. Quality control of the resulting libraries was carried out using the DNA High Sensitivity Bioanalyzer Reagents (Agilent Technologies) and concentrations were measured via Qubit HS DNA Quantification Kit (Thermo Fisher). Sample pools were then seeded onto UG sequencing beads, pre-enriched, and amplified by emulsion PCR, leveraging UG's automated sequencing bead preparation system (AMP). Sequencing was performed on UG100 sequencing systems, running 464 flow-cycles (116 cycles across each of the four nucleotides [T, G, C, A]).

## Ultima genomics sequencing analysis

We aligned the reads to the human reference genome hg38[23] with STAR version 2.7.10b[58]. We sorted the resulting.bam files utilizing samtools version 1.16.1[57]. We thereafter processed these alignments with the subread module featureCounts version 2.0.1[24] to verify de novo transcript expression, and subsequently extracted normalized counts for correlative analysis with SARTools version 1.8.1[25], a package wrapping DESeq2 version 1.36.0[18].

## Reporting summary

Further information on research design is available in the Nature Portfolio Reporting Summary linked to this article.

## Data availability

Datasets have been uploaded to the NASA Open Science Data Repository (OSDR; osdr.nasa.gov, accession number OSD-569) and made publicly accessible. Processed data are available at the same address.Related data can be found in the parallel SOMA papers[59–69].

## Code availability

Scripts utilized for the presented analysis have been deposited in the following repository: https://github.com/eliah-o/inspiration4-omics/tree/main/i4_direct_rna.

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

## Acknowledgements

C.E.M. thanks the WorldQuant Foundation, the GI Research Foundation, Katherine & John Bleckman Foundation, NASA (NNX14AH50G, NNX17AB26G, 80NSSC22K0254, NNH18ZTT001N-FG2, 80NSSC23K0832), the National Institutes of Health (R01MH117406, P01CA214274 R01CA249054, R01AI151059), and the LLS (MCL7001-18, LLS 9238-16). J.K. thanks MOGAM Science Foundation. J.K. was supported by Basic Science Research Program through the National Research Foundation of Korea (NRF) funded by the Ministry of Education (RS-2023-00241586). J.K. acknowledges Boryung for their financial support and research enhancement ground, provided through their Global Space Healthcare Initiative, Humans In Space, including mentorship and access to relevant expert networks. This research was supported in part by the Intramural Research Program of the National Library of Medicine. We also would like to thank Ultima Genomics for the deep RNA-seq profiles and collaboration.

## Author contributions

C.E.M. and J.M. conceived the study. D.N., N.D., and K.A.R. generated the sequencing data. K.G., T.M.N., E.O., N.H., J.K., and C.E.M. analyzed and interpreted the data and wrote the manuscript. E.E.A., J.T., D.T., and A.M.M. revised the manuscript.

## Competing interests

C.E.M. is a co-founder of Cosmica Biosciences. The remaining authors declare no competing interests.
