## [Peer Review File · Nature Communications]

Direct RNA sequencing of astronauts reveals
spaceflight-associated epitranscriptome changes and
stress-related transcriptional responsesREVIEWER COMMENTS

Reviewer #1 (Remarks to the Author):

This pioneering study utilized ONT DRS to investigate the effects of spaceflight on human physiology at the epitranscriptome level. The authors analyzed DRS profiles and identified key genetic pathways that exhibited significant changes in four SpaceX Inspiration4 astronauts before spaceflight, upon return, and during recovery. Significantly, this is the first m6A methylation profile for a human space mission. The authors demonstrated that ONT could be a robust and efficient platform to monitor the molecular dynamics of astronauts during a flight mission. This study represents just the beginning of an accumulation of information that will lead to greater understanding of the human body and increased confidence in dealing with deep space travel in the future. Technically, the authors performed several routine analyses to identify changes in the transcriptome and epitranscriptome. However, I have some minor questions and suggestions for the authors:

1. Can the authors provide a general comparison of their results with those of previous studies that used other sequencing platforms to investigate gene expression in astronauts?
2. In Table 1, the authors found several unique transcripts at one timepoint. Is it possible that these genes were caused by low sequencing depth? Please clarify.
3. To illustrate the exemplified genes with changes, please provide an IGV snapshot or a visualizable gene structure and sequencing results in another form.
4. The authors first used m6anet to identify the methylated sites, and then applied the methylKit tool, which was designed to analyze DNA methylation, to find the differentially methylated m6A profile. This appears unusual. I would suggest that they use another specialized tool to directly call diff-m6A. At the very least, they should compare results from different tools.
5. ONT has its own advantages over other sequencing platforms, especially in space. Can the authors expand their imagination and speculate on the future scenario of instant use of ONT in space?

Reviewer #2 (Remarks to the Author):

The investigators provide an transcriptomic analysis of samples collected from the I4 astronauts both before, immediately after and during recovery from their space flight. This is a unique and valuable resource that is here studied using a novel and interesting technology (long-read direct RNA sequencing on the ONT nanopore platform). Despite the obvious interest and impact of the study design, there are a few issues confounding a rigorous analysis and interpretation of the data, as described below that greatly limit the scope of this report. Together with a somewhat superficial scrutiny of the output data, this report appears to be somewhat preliminary.

Major Comments:

A pervading issue in this report is in how individual datasets are treated during statistical analyses. It is not clearly stated, but I assume each individual is treated as a 'biological' replicate, thus yielding $n=4$ for most samples. If technical replicates were obtained in this study, it is not clearly described. Some clarity on whether this process would satisfy the assumptions made by the underlying statistical tests is warranted. Related to this, given the small size of the cohort (4 individuals) some discussion of how inter-person variability (e.g. age, genetic background, biological sex) might impact these studies since there is also likely to be considerable intra-subject variability.

Similarly, since the RNA studied is derived from whole-blood collections, changes in the astronaut blood chemistry/cell counts/etc is likely to be very different between individuals and may skew the abundance of genes/gene sets found for each individual. For example, should one individual exhibit mild leukocytosis after the space flight, then more RNA transcripts/gene sets associated with white

blood cells would be found, rather than any actual changes in the expression profiles of these genes. As a result, without supporting information and/or characterization of the samples collected and tested, the investigators cannot distinguish between changes in transcriptomic/epitranscriptomic profiles due to gene expression regulation, versus capture of different transcriptomic/epitranscriptomic features due to changes in the complexity/compositions of the samples being studied.

This study is limited by the lack of any in vitro validation of the observed genes in gene expression and/or isoform abundance. While RT-PCR validation of differentially expressed genes is not necessarily required since this is a pretty well established technology; the small number of novel isoforms, including novel isoforms with putative retain introns (for example) warrants follow-up studies (if there are biological materials still remaining). This point is particularly important given the short length of many of the 'long-reads' collected in this study and that the unique technology used in this study is long-read sequencing. Incomplete capture of full-length transcripts in the direct RNA sequencing protocol may well confound the interpretation and robustness of the downstream computational analysis (as eluded to in the discussion).

Minor Comments:

The investigators do not provide a concise description of the samples collected or study design. While this information can be gleaned from the supplementary tables and may well be described elsewhere in other associated manuscripts, some text describing the basic cohort, timing of collections, type of samples collections, number of replicates, etc, etc would be helpful, even if this information is provided elsewhere. For example 'R+1' is not defined (although it is quite obvious what this would be, it would be good to provide this information directly). Similarly, how the current study is distinct from the Overbey manuscript (i.e. different samples/different technological focus) would be important, since it appears that this is a companion paper.

Also related to this, since this is a ONT-focused manuscript, some descriptions of the datasets collected would be appropriate before launching into the results of the data analysis. E.g. number of reads, number of mapped reads, length of reads, etc, etc. Again, although this information might be gleaned from supplementary materials, some descriptive statements on the data collected and data quality and how these compare to other publicly accessible datasets or standards would be helpful.

Some discussion/comments on why there are numerous genes that show no DE with the SARTools pipelines but do show changes with the ONT pipeline would be valuable (i.e. the red points along the x=0 axis). Is this due to the multiple mapping allowed by the ONT pipeline?

The vertical text in Figure 2B is hard to read and some definitions are missing. For example, "NEBEN_AML_WITH_FLT3_OR_NRAS_UP" is ill-defined. While I appreciate there are many pathways to be considered, defining the top few (as depicted in Figure 2B) with intuitive labels would be useful.

In figure 2A and 2B, without any labels for what each row corresponds to, the images are impossible to interpret or scrutinize, other than to provide a broad overview of some (correlated?) changes in the expression levels or some genes amongst the cohort.

For the section describing the m6A analysis, some high-level description of the number of putative m6A sites identified in each dataset and how many genes/transcripts show putative changes, and how many of these correlate with the DE analysis performed in the previous section is needed. Some of this information is found in Figure 4, but could be expounded upon.

When comparing R+1 to all pre-flight timepoints, is each timepoint from each individual treated as though they are independent replicates? This should be clearly stated in the maintext, in addition to

the methods.

In figures 3 and 4, much of the small text is difficult to read and should be reworked for clarity.

In the discussion, the authors state "LRS can precisely identify such differences, given that the nearly all 257 transcripts are shorter than the theoretical maximum sequencing length for Nanopore platforms, which extends to nearly 2.3Mb". This statement is somewhat mis-leading, since the major limitations to capturing full-length RNA (or cDNA) transcripts are factors such as RNA fragmentation/integrity, success of long-range reverse-transcription (often limited to 4-5kb by standard RTs) and the quality of the final library loaded onto a flowcell.

Reviewer #3 (Remarks to the Author):

The manuscript from Grigorev K. et al describes the transcriptome and m6A methylome of 4 astronauts on land, in space, and after returning. While this is an invaluable dataset that has incredible potential to be interesting, unfortunately, the current way that the manuscript is presented makes it incredibly boring. I appreciate that the authors were rigorous in using different methods to analyze the data, however, they did not seem to be able to extract much meaningful biology out of this dataset, diminishing my interest in this manuscript.

Here are my major comments:

- 1) What is the distribution of the biotype of the genes that are disrupted upon flight- are they mostly coding, or are there also noncoding RNAs? How many transcripts did the authors identify?
- 2) What percentage of the transcripts that showed expression changes during flight were recovered postflight? Are there transcripts that recovered faster versus slower?
- 3) Are there shared regulators (such as RBPs) for the transcripts that recover or show similar expression changes?
- 4) For the m6A differential sites- what is the distribution of their changes along a transcript?
- 5) For the genes with differential m6A modifications- do they show gene expression changes?
- 6) Are the m6A modifications recovered post-flight?
- 7) For the new transcript that the authors identified- can they show the read count overlaid into the UCSC genome browser to enable the readers to get a sense of the transcript and its neighbourhood. Is the gene conserved in other organisms?

Dear Reviewers and the editors,

We thank the reviewers for their careful review of our study and results, and we have updated our manuscript with updated figures, improved text, and additional validation data sets. We have responded to their questions below, with reviewer comments in black and our responses in blue.

Reviewer #1 (Remarks to the Author):

This pioneering study utilized ONT DRS to investigate the effects of spaceflight on human physiology at the epitranscriptome level. The authors analyzed DRS profiles and identified key genetic pathways that exhibited significant changes in four SpaceX Inspiration4 astronauts before spaceflight, upon return, and during recovery. Significantly, this is the first m6A methylation profile for a human space mission. The authors demonstrated that ONT could be a robust and efficient platform to monitor the molecular dynamics of astronauts during a flight mission. This study represents just the beginning of an accumulation of information that will lead to greater understanding of the human body and increased confidence in dealing with deep space travel in the future. Technically, the authors performed several routine analyses to identify changes in the transcriptome and epitranscriptome. However, I have some minor questions and suggestions for the authors:

We thank Reviewer #1 for recognizing the significance of this study and providing helpful feedback. We agree that this dataset provides the first-ever view of such data, and as such, will serve as a baseline data set for a deeper understanding of human physiological and transcriptome effects due to spaceflight. Please find attached our point-by-point responses and updates to the manuscript.

1. Can the authors provide a general comparison of their results with those of previous studies that used other sequencing platforms to investigate gene expression in astronauts?

Yes, we have now updated this section in the Introduction and Discussion, and we also provide a broader review of other studies. To comprehensively detail previous studies that investigated gene expression in astronauts, we reviewed the NASA GeneLab human transcriptomic studies and placed the following revised summary of previous studies in our introduction: “Ground studies collecting transcriptomic profiles of human subjects have attempted to characterize the impact of individual components of spaceflight, such as radiation or microgravity exposure and simulated with a parabolic flight. With regard to direct spaceflight exposure, the available transcriptomic data is even more limited, focused on a few specific assays, either sequencing the human IgM repertoire following long-term spaceflight, measuring the proteome of exhaled breath condensate after long-term spaceflight, or examining gene expression profiles from astronaut hair follicle samples after long stays on the International Space Station. However, beyond this limited work, there is scant data on the impact of human spaceflight on the transcriptome beyond the NASA Twins Study (1-year mission), and almost no data from short-duration missions. Moreover, there are no single-molecule, direct RNA datasets from

astronauts, which can help detail changes in RNA methylation and base modifications while also capturing expression and isoform changes.”

2. In Table 1, the authors found several unique transcripts at one time point. Is it possible that these genes were caused by low sequencing depth? Please clarify.

To mitigate any low-coverage issues with these data, we have created a minimum coverage threshold for each gene, as suggested by the authors of the *StringTie* program. On a per-sample basis, there exists some variability in coverage, with most samples mapping between 1 and 2 million LRS reads. These differences become less pronounced when comparing between time points. Therefore, in order to extract meaningful biological signals, we required that transcripts be shared among all samples for a given time point, where some samples had below-average coverage and some had above-average coverage, and that they not be shared by all other samples within the comparison set, spanning a similar diverse range of coverages. Given this analytical framework, we believe that the analysis is robust to differences in sequencing depth. We have additionally updated Supplemental Table S5 to provide metrics for per-sample read alignment and coverage. We now also include orthogonal validation data from a distinct NGS platform (Ultima UG100).

3. To illustrate the exemplified genes with changes, please provide an IGV snapshot or a visualizable gene structure and sequencing results in another form.

We have added Supplemental Figure S3, which displays the raw alignments informing this novel transcript variant and demonstrates its raw base pair sequence. Additionally, we have published and included a link to a UCSC Genome Browser Session for viewers interested in integrating the data for this particular gene for all samples with a wide variety genome-wide analyses.

4. The authors first used m6anet to identify the methylated sites, and then applied the methylKit tool, which was designed to analyze DNA methylation, to find the differentially methylated m6A profile. This appears unusual. I would suggest that they use another specialized tool to directly call diff-m6A. At the very least, they should compare results from different tools.

Since the submission of our article, a benchmarking study on m⁶A callers was published, “*Systematic comparison of tools used for m⁶A mapping from nanopore direct RNA sequencing*” (10.1038/s41467-023-37596-5), which demonstrates that m6anet is the most accurate m6a caller, and we have now cited in the Discussion: “While accounting for secondary mappings with *salmon* could be argued to be the more sound approach, this calls for further assessment and discussion of methods to be used for analyses of direct-RNA data, as retention of secondary mappings is incompatible with m⁶A callers such as *m6anet*, which was recently shown to be the most accurate m⁶A caller within a comparative software study.” Once m⁶A modifications have been called, there are even fewer standardized parameters for the downstream analysis of RNA modifications. We selected methylKit to call differences between the given probabilities, since it employs Fisher’s exact test, which is a suitable, non-parametric test for comparing groups with

small sample sizes. We lastly point to one prior publication which serves as precedent for employing this particular combination (10.1101/2022.01.24.477497).

In our estimation, m6anet was the best suited for our particular experimental design and sequencing data, since many other m⁶A callers, including nanocompore, differr, DRUMMER, JACUSA2, xPore, and yanocomp, require a negative control, low-m⁶A sample, or knock-out model. Of note, m6anet does not require a low-m⁶a profile because it is applying a machine learning model which is able to generally differentiate electrical signals corresponding to a modified base pair from an unmodified base pair. There are other similarly designed tools, including ELIGOS, EpiNano, nanom⁶A; however, they were trained on data from older chemistries and base-calling tools. We used m6anet since it does not have this limitation.

5. ONT has its own advantages over other sequencing platforms, especially in space. Can the authors expand their imagination and speculate on the future scenario of instant use of ONT in space?

We agree that this is a major advantage to the ONT platform and we have now edited our discussion to emphasize this (novel text in *italics*), with the first paragraph reading as follows: “All prior data from spaceflight studies have used short-read sequencing. The advent of direct-RNA sequencing in particular allows for additional intrinsic information about the RNA, such as m⁶A methylation, to be detected.

“The development of space-specific, microgravity-compatible sequencing protocols foreshadows the exciting possibility of real-time monitoring of changes of the transcriptome and epitranscriptome for future missions.”

We additionally discuss the experimental demonstration of direct-RNA sequencing, the same as in this paper, within the final paragraph of the discussion:

“Finally, given the recently proposed missions with private (SpaceX, Axiom, Sierra Space) and public (NASA, ESA, and JAXA) space entities, there will be continued opportunities to replicate these findings and to continue to discover new features of the transcriptome. This will include continued mapping of space-related gene expression responses, m⁶A site changes, and isoform-switching events that are associated with spaceflight. Also, in-flight testing of direct RNA sequencing has been demonstrated on the ISS, which indicates that LRS of RNA could also be applied for crew studies in future missions, including for blood, microbial, and environmental samples. Optimization and deployment of these single molecule technologies can also aid in the plans for lunar laboratories, in-flight clinical diagnostics, and the ability to discover non-canonical bases or new base modifications for lunar and exploration-class (e.g. Mars) missions.”

On a final, exciting note, we have just completed flight testing of the Mk1C device at SpaceX headquarters, which is being QC'd for upcoming missions and should hopefully fly to space within a year, and will enable us to test these protocols directly in flight.

Reviewer #2 (Remarks to the Author):

The investigators provide an transcriptomic analysis of samples collected from the I4 astronauts both before, immediately after and during recovery from their space flight. This is a unique and valuable resource that is here studied using a novel and interesting technology (long-read direct RNA sequencing on the ONT nanopore platform). Despite the obvious interest and impact of the study design, there are a few issues confounding a rigorous analysis and interpretation of the data, as described below that greatly limit the scope of this report. Together with a somewhat superficial scrutiny of the output data, this report appears to be somewhat preliminary.

We thank Reviewer #2 for their interest in this study and their informed comments. Throughout our analysis, we wished to present the results in a manner which did not overgeneralize, given limitations in terms of astronaut numbers. We aimed simply to provide a series of robust observations (especially changes observed in all crew members and with multiple pipelines), providing a useful roadmap for future transcriptomic analysis, as well as highlighting key genes and pathways, which can guide future spaceflight missions and experiments. In line with this general statement, we attach our point-by-point responses, and also highlight new data and analyses in our revision.

Major Comments:

A pervading issue in this report is in how individual datasets are treated during statistical analyses. It is not clearly stated, but I assume each individual is treated as a 'biological' replicate, thus yielding $n=4$ for most samples. If technical replicates were obtained in this study, it is not clearly described. Some clarity on whether this process would satisfy the assumptions made by the underlying statistical tests is warranted. Related to this, given the small size of the cohort (4 individuals) some discussion of how inter-person variability (e.g. age, genetic background, biological sex) might impact these studies since there is also likely to be considerable intra-subject variability.

Indeed we used 4 biological replicates for each time point, and then compared between groups. We have clarified this within the manuscript by adding the following information to the presentation of **Supplemental Table S1**: "Both pipelines were applied to the data in the context of multiple flight, recovery, and longitudinal profiles (**Supplemental Table S1**). A single sequencing run was performed for each astronaut and time-point; the four astronauts were grouped together as biological replicates for downstream analysis." Ultimately, technical replicates are not an input recommendation or requirement for *edgeR*, *DESeq2* or *methyKit*.

As we have processed the data, in accordance with best practices, we have only utilized de-identified data. Therefore, we cannot specifically associate observed inter-person variability with the personal health information of the astronauts themselves. Of note, the flight subjects were considered as covariates in the differential expression analyses and linear models, to account for inter-person variability. We have clarified this within the Methods section of the manuscript: "Covariates to account for inter-astronaut variability were included when performing both *DESeq2* and *edgeR* analysis."

Also, we have added a section in the Discussion about this important point: “Moreover, given the civilian background of the I4 astronauts, we note that transcriptomic changes can be confounded with age, biological sex, preflight preparation differences, or divergent postflight recovery procedures, and thus utilized tools that aim to model and normalize such variation.”

Similarly, since the RNA studied is derived from whole-blood collections, changes in the astronaut blood chemistry/cell counts/etc is likely to be very different between individuals and may skew the abundance of genes/gene sets found for each individual. For example, should one individual exhibit mild leukocytosis after the space flight, then more RNA transcripts/gene sets associated with white blood cells would be found, rather than any actual changes in the expression profiles of these genes. As a result, without supporting information and/or characterization of the samples collected and tested, the investigators cannot distinguish between changes in transcriptomic/epitranscriptomic profiles due to gene expression regulation, versus capture of different transcriptomic/epitranscriptomic features due to changes in the complexity/compositions of the samples being studied.

This is an important consideration, and we now include here results included in a companion paper (Kim et al) that used FACS to characterize cell proportions from the mission. We also now include the complete blood count (CBC) and comprehensive metabolic panel (CMP) results from the CLIA lab (Quest) blood processing that was performed on the crew, which detailed the proportion of cell types found in the blood at all time points. Both methods showed no significant difference (Wilcoxon Rank Sum, p-values all >0.1) in the proportion of the cell types in the blood in pre- and post-flight, which should minimize the impact of cell type representation in the blood transcriptome data.

Reviewer Figure 1. PBMC subpopulation markers for i4 crew samples confirmed with FACS. (left) FACS gating of T cell, B cell, NK cell, and monocytes from the i4 PBMCs and (right) Cell proportion of i4 PBMC over time calculated from FACS, with non-significant (NS) results for the proportions of cell types before and after spaceflight.

a**Complete Blood Count****b****Comprehensive Metabolic Panel**
Reviewer Figure 2. CBC and CMP profiles of the i4 crew before and after spaceflight. Each dot represents each crew member (different colored dots for C001, red, C002, green, C003, blue, C004, purple, listed on the legend on the right side) for the Complete Blood Count (CBC) profile (top) and Comprehensive Metabolic Panel (CMP, bottom) panel, with cell types and analytes annotated on top of each plot. Measured analytes were all within nominal range (white zones) or within 5-10% of the nominal range.

This study is limited by the lack of any in vitro validation of the observed genes in gene expression and/or isoform abundance. While RT-PCR validation of differentially expressed genes is not necessarily required since this is a pretty well established technology; the small number of novel isoforms, including novel isoforms with putative retain introns (for example) warrants follow-up studies (if there are biological materials still remaining). This point is particularly important given the short length of many of the 'long-reads' collected in this study and that the unique technology used in this study is long-read sequencing. Incomplete capture of full-length transcripts in the direct RNA sequencing protocol may well confound the interpretation and robustness of the downstream computational analysis (as eluded to in the discussion).

We have verified the existence of *de novo* transcript variants within the Ultima Genomics deep-RNA sequencing platform, verifying expression within the embedded genomic loci. We verified the expression of 23 out of 25 (92%) transcripts utilizing this technology, which provides a strong basis for their validity and is, by itself, an in-depth RNA-seq data set for future analysis and studies.

We agree that future samples characterization will be important to understanding these RNA changes' physiological relevance for spaceflight. Given the nature of *de novo* transcriptome analysis, we detected specific reads which were unique to a specific time-point across astronauts; however, we made no speculation as to the physiological relevance of these findings. Therefore, among the 25 different transcripts identified in **Table 1**, we can only use predicted functions. We plan to biobank all remaining biological material, to ensure its availability for future comparative studies. Indeed, we hope to replicate similar trends via *de novo* transcriptome and deep RNA sequencing analysis for the Polaris Dawn mission (<https://polarisprogram.com/dawn/>), and thereafter return to genes of interest appearing across two separate astronaut missions into low-earth orbit.

Minor Comments:

The investigators do not provide a concise description of the samples collected or study design. While this information can be gleaned from the supplementary tables and may well be described elsewhere in other associated manuscripts, some text describing the basic cohort, timing of collections, type of samples collections, number of replicates, etc, etc would be helpful, even if this information is provided elsewhere. For example 'R+1' is not defined (although it is quite obvious what this would be, it would be good to provide this information directly). Similarly, how the current study is distinct from the Overbey manuscript (i.e. different samples/different technological focus) would be important, since it appears that this is a companion paper.

Thank you for this comment. We have now posted a complete and in-depth description of the protocols used in this paper, as well as the connection to other assays and samples, in a separate submission and companion paper, "Collection of Biospecimens from the Inspiration4 Mission Establishes the Standards for the Space Omics and Medical Atlas (SOMA)."

<https://www.ncbi.nlm.nih.gov/pmc/articles/PMC10187258/>. We have clarified this connection at the top of the **Methods section**: “Methods // **Direct RNA sequencing**. Total RNA was processed and sequenced as described in Collection of Biospecimens from the Inspiration4 Mission Establishes the Standards for the Space Omics and Medical Atlas (SOMA).”

Moreover, we have updated the terminology (e.g. R+1) and clarified the text. Specifically, we have included the following sentence in the **Introduction**: “We here analyze LRS samples taken before (L-92, L-44, L-3 days) and after (R+1, R+45, R+82, R+194 days) spaceflight.” As previously mentioned, we have clarified the treatment of replicates by adding the following information to the presentation of **Supplemental Table S1**: “Both pipelines were applied to the data in the context of multiple flight, recovery, and longitudinal profiles (**Supplemental Table S1**). A single sequencing run was performed for each astronaut and time-point; the four astronauts were grouped together as biological replicates for downstream analysis.”

Also related to this, since this is a ONT-focused manuscript, some descriptions of the datasets collected would be appropriate before launching into the results of the data analysis. E.g. number of reads, number of mapped reads, length of reads, etc, etc. Again, although this information might be gleaned from supplementary materials, some descriptive statements on the data collected and data quality and how these compare to other publicly accessible datasets or standards would be helpful.

These are also important points, and we have updated the manuscript to help provide clarity. We have also added more details to the text and the supplemental tables, to help clarify these points. Specifically, **Supplemental Table S5** contains information regarding the number of mapped reads, transcript sensitivity and coverage. Based on this, we have specifically included the following sentence at the beginning of the results section: “All runs exhibited a minimum of 10-times transcriptome coverage post-alignment.” We include information regarding gene lengths within the **Methods** section: “Additionally, the sub-2Kbp read lengths (median length between 517 and 1,331, with one outlier of 279, Supplemental File S1) were insufficient to perform differential transcript usage analyses, and we focused on the differential gene expression results, which pipeline-transcriptome-de generates with edgeR while collapsing transcripts into respective genes.”

The relationship between the number of total reads and mapped reads is based on both stochastic chemical sequencing processes and initial RNA concentration input; we have verified this by replacing the reference genome with the Nanopore RNA calibration strand FASTA file, showing that nearly all unmapped reads map to this sequence, which is included by default within the Nanopore Direct RNA Sequencing kit, SQK-RNA002 (<https://help.nanoporetech.com/en/articles/6632031-what-is-rna-cs-racs>).

Given the novelty of the technique, there are few publications related to particular direct-RNA standards. More than 80% of publicly available datasets do not include deposition of raw FAST5 files, which limits the potential to perform a meta-analysis of data quality across publicly available datasets. There are approximately 500 samples deposited within the European

Nucleotide Archive (ENA) which include fast5 files; however, we believe that a meta-analysis to check for data quality is outside of the scope of this study. Since we have implemented the most recently available sequencing chemistry and basecalling methodology, we are confident that our technical data quality matches current best-practices. The basecalling metrics included in Supplemental File S1 suggest excellent Nanopore base-calling quality, according to manufacturer's recommendations and ENA best practices.

Some discussion/comments on why there are numerous genes that show no DE with the SARTools pipelines but do show changes with the ONT pipeline would be valuable (i.e. the red points along the x=0 axis). Is this due to the multiple mapping allowed by the ONT pipeline?

We have commented now that such differences are likely due to the heuristics in the algorithms, rather than multi-mapping reads, and this is clarified in the text. Specifically, we have addressed this in the Discussion: "Additionally, the associated differential expression callers, *DESeq2* and *edgeR*, respectively, utilize slightly different heuristics to perform \log_2 fold change estimates, resulting in many genes uniquely identified by the *salmon-edgeR* pipeline mapping to the y-axis of Figure 1b."

The vertical text in Figure 2B is hard to read and some definitions are missing. For example, "NEBEN_AML_WITH_FLT3_OR_NRAS_UP" is ill-defined. While I appreciate there are many pathways to be considered, defining the top few (as depicted in Figure 2B) with intuitive labels would be useful.

Thank you - we have updated the Figure for readability. Of note, the pathway names we used are exactly as they appear in the MSigDb database, to enable easier comparison with other datasets. In order to ensure the reproducibility of the figure, we need to match the name with the original database identifier. Nevertheless, we have added the following sentence to each figure legend referencing these labels: "Gene sets can be explored further at: <https://www.gsea-msigdb.org/gsea/msigdb>." In this manner, researchers can navigate to the Broad Institute page and utilize the search feature to better understand the internal features of these gene sets. For the case of "NEBEN_AML_WITH_FLT3_OR_NRAS_UP", the relevant page would be: https://www.gsea-msigdb.org/gsea/msigdb/human/geneset/NEBEN_AML_WITH_FLT3_OR_NRAS_UP.html?keywords=NEBEN_AML_WITH_FLT3_OR_NRAS_UP

In figure 2A and 2B, without any labels for what each row corresponds to, the images are impossible to interpret or scrutinize, other than to provide a broad overview of some (correlated?) changes in the expression levels or some genes amongst the cohort.

We have updated these figures. Specifically, we have added gene names to the heatmap, providing labels for each row.

For the section describing the m6A analysis, some high-level description of the number of putative m6A sites identified in each dataset and how many genes/transcripts show putative

changes, and how many of these correlate with the DE analysis performed in the previous section is needed. Some of this information is found in Figure 4, but could be expounded upon.

We have incorporated these statistics into the manuscript in the following paragraph: “Next, we identified the sites of m⁶A modification across isoforms and their degree of change, leveraging the single molecular nature of the ONT platform to find sites differentially methylated for the flight, recovery, and longitudinal profiles (q-value < 0.01, *methylKit*³⁰) (Supplemental Table S4). Herein, we recorded 1,190 m⁶A modification sites that demonstrated a statistically significant difference for at least one particular longitudinal comparison. These sites were located on 200 total transcripts, sourced from 193 gene loci.”

Comparing the 193 gene loci which contained DRACH motifs with differentially methylated sites (q-value < 0.01, *methylKit*) with the set of 440 differentially expressed gene loci (FDR-adjusted p-value < 0.05, *either salmon or SARTools*), we find a modest overlap of 53 genes. The relationship between gene expression and m⁶a methylation is still being examined, although it is likely a key regulator of mRNA decay. Within our analysis, we postulated on the functional implications of these shared genes by comparing gene sets, as opposed to individual genes.

When comparing R+1 to all pre-flight timepoints, is each timepoint from each individual treated as though they are independent replicates? This should be clearly stated in the maintext, in addition to the methods.

This has been updated in the methods; thank you for the careful reading. Specifically, we have added the following sentence to the **Differential Gene Expression Analysis** methods section: “For comparisons where multiple time-points were grouped as either preflight or postflight, each sequence run was treated as an independent replicate.”

In figures 3 and 4, much of the small text is difficult to read and should be reworked for clarity.

We have updated these figures to improve clarity. Thank you.

In the discussion, the authors state "LRS can precisely identify such differences, given that the nearly all 257 transcripts are shorter than the theoretical maximum sequencing length for Nanopore platforms, which extends to nearly 2.3Mb". This statement is somewhat mis-leading, since the major limitations to capturing full-length RNA (or cDNA) transcripts are factors such as RNA fragmentation/integrity, success of long-range reverse-transcription (often limited to 4-5kb by standard RTs) and the quality of the final library loaded onto a flowcell.

We thank the reviewer for highlighting this statement; we agree that it requires additional context. We have altered the relevant section to read as follows: “LRS can potentially identify all such differences, given that nearly all transcripts are shorter than the theoretical maximum sequencing length for Nanopore platforms (2.3Mb). Nevertheless, limitations remain within the library preparation and technical specifications (e.g. 3' enrichment) of different LRS platforms,

and improvements in these protocols and informatics methods will help profiling for future samples.”

In terms of the specific issues mentioned above, we believe that these are generally applicable to all kinds of sequencing approaches, including second generation short-read sequencing and third-generation long-read sequencing. In the context of our study, long-range reverse-transcription is an optional component of the sequencing protocol, and therefore not a major concern in terms of transcript length. We believe that we have identified and expanded upon the major technical bias confounding transcript-level analysis within the discussion paragraph regarding 5' end truncation. Reference 44, which cites an article entitled *Opportunities and challenges in long-read sequencing data analysis* (10.1186/s13059-020-1935-5) provides in our estimation an excellent review of technical bias as they apply for all commercially available long-read sequencing platforms.

Reviewer #3 (Remarks to the Author):

The manuscript from Grigorev K. et al describes the transcriptome and m6A methylome of 4 astronauts on land, in space, and after returning. While this is an invaluable dataset that has incredible potential to be interesting, unfortunately, the current way that the manuscript is presented makes it incredibly boring. I appreciate that the authors were rigorous in using different methods to analyze the data, however, they did not seem to be able to extract much meaningful biology out of this dataset.

We thank Reviewer #3 for recognizing the significant interest in this dataset. Our first draft was perhaps too cautious in our interpretation; nevertheless, we hope that the mentioned rigorous analytical approach will provide grounds for robust biological interpretation, especially when this data is placed in context of future similar spaceflight observational studies, and in conjunction with the proliferation of commercial spaceflight efforts. Moreover, we have updated the discussion to highlight the significance of the study and the importance of the R+1 observations at the gene and first-ever observation of m⁶A level dynamics in astronauts.

Here are my major comments:

1) What is the distribution of the biotype of the genes that are disrupted upon flight- are they mostly coding, or are there also noncoding RNAs? How many transcripts did the authors identify?

We have clarified this in the text, and below, to note that almost all genes were protein-coding. Within the time-points we believed fit the description of demonstrating “disruption upon flight” - FP1, RP1, RP2, RP4, RP5 and RP6, further defined in Supplemental Table S1 - we noted that there were 130 transcripts that were differentially expressed. Based on the technical specifications of the Nanopore Direct RNA-sequencing protocol (SQK-RNA002), only poly-A tailed RNAs are robustly captured by the assay. We therefore observe differential expression of primarily protein-coding genes (green) as shown in the figure below.

Reviewer Figure 3. Gene annotation of types of genes showing expression and methylation changes by timepoint. Each gene (y-axis) is annotated by the time point of differential change (column) and gene type (gene= protein-coding, orange= rRNA, brown=mitochondrial, blue=lncRNA, pink=pseudogene, light-green=snRNA, yellow=scaRNA).

2) What percentage of the transcripts that showed expression changes during flight were recovered postflight? Are there transcripts that recovered faster versus slower?

We observed that approximately 90% of transcripts that demonstrated expression changes were recovered postflight. Given the short duration of the actual low-earth orbital spaceflight, we saw that all transcripts were recovered after 45 days, which was the first sampling time post spaceflight. Therefore, we did not have the resolution to examine the actual time velocity vector for transcript recovery. Ultimately, we analyzed the functional implications of these gene changes within the context of pathway analysis, as further described in Figure 3.

3) Are there shared regulators (such as RBPs) for the transcripts that recover or show similar expression changes?

We thank the reviewer for this suggestion. We chose to utilize ChEA3 to answer this question (<https://maayanlab.cloud/chea3/>), since it integrates information from fairly recent CHIP-seq, RNA-seq and gene set databases to generate a holistic score. Within the manuscript, we have added the following section into the Results section: "Transcription factor enrichment analysis was then utilized to identify commonly shared transcription factors by genes (Supplemental Table S2) across all time points. The top three suggested regulators, based on a combined CHIP-seq and co-expression analysis, KLF1, GATA1, and TAL1 are all key transcription factors (TFs) for erythrocyte differentiation^{27–29}, supporting the changes in these cell types and also suggesting possible TF-based drivers for the trends observed within the GESECA analysis (Supplemental Table S3)." Likewise, we have added the following section to our Methods section: "**Transcription factor enrichment analysis.** Gene annotations within **Supplemental Table S2** were converted to gene names utilizing gProfiler (<https://biit.cs.ut.ee/gprofiler/convert>).⁵¹ The resulting gene list was input into the ChEA3 digital web server (<https://maayanlab.cloud/chea3/>).⁵² The mean rank results were exported and reported in **Supplemental Table S3**." In our experience, programs which predict regulation are heavily reliant on their particular source of data. For example, programs such as BART (<https://zanglab.github.io/bart/>) and ChIP-Atlas (https://chip-atlas.org/enrichment_analysis), provide regulatory profiles highly dependent on the contexts of the sourced CHIP-seq data, which may not be applicable to spaceflight contexts, which is another reason we used ChEA3.

4) For the m6A differential sites- what is the distribution of their changes along a transcript?

For readers to better examine questions related to the differential m6A profiles, we have updated **Supplemental Table S4** to include the definitions for the thirteen different comparisons contained within, noting redundancy with **Supplemental Table S1**. *m6anet*, the m6a modification quantifier for Nanopore Direct RNA-sequencing data, was utilized to obtain per-site methylation probabilities. *m6anet* only makes predictions on transcript positions with DRACH motifs, which encompass the predominant consensus motifs for m6a methylation in humans. Each DRACH motif needs at least 20 reads of coverage in order to be eligible for a prediction. Specific predicted DRACH motif positions were included in **Supplemental Table S4** if their *methylKit* q-value fell below 0.01 for at least one comparison.

Based on these criteria, we identified 1,190 DRACH motifs that demonstrated a statistically significant difference in one particular comparison time-point. These DRACH motifs were located on 200 total transcripts, sourced from 193 gene loci. However, since our long-read data exhibited 5' truncation, due to polyA-tail priming, we ultimately chose to consider these m⁶A differences as discrete units; either in terms of transcript location or transcript identity. Unfortunately, assessing for differences within different gene segments, such as the 5' UTR, CDS and 3' UTR for protein-coding genes, will likely reveal changes caused by technical truncation, as a function of the transcript length, as opposed to true biological differences.

We have incorporated these specific statistics into the manuscript in the following paragraph: "Next, we identified the sites of m⁶A modification across isoforms and their degree of change, leveraging the single molecular nature of the ONT platform to find sites differentially methylated for the flight, recovery, and longitudinal profiles (q-value < 0.01, *methylKit*³⁰) (**Supplemental Table S4**; OSDR accession number OSD-569). Herein, we recorded 1,190 m⁶A modification sites that demonstrated a statistically significant difference for at least one particular longitudinal comparison. These sites were located on 200 total transcripts, sourced from 193 gene loci."

5) For the genes with differential m⁶A modifications- do they show gene expression changes?

Comparing the 193 gene loci which contained DRACH motifs with differentially methylated sites (q-value < 0.01, *methylKit*) with the set of 440 differentially expressed gene loci (FDR-adjusted p-value < 0.05, *either salmon or SARTools*), we find a modest overlap of 53 genes. The relationship between gene expression and m⁶A methylation is still being examined, although it is likely a key regulator of mRNA decay. Within our analysis, we postulated on the functional implications of these shared genes by comparing gene sets, as opposed to individual genes. This has also been updated in the manuscript.

6) Are the m⁶A modifications recovered post-flight?

This is a great question, which we have now added to our manuscript. To answer this question, we examined the m⁶A differences observed within the FP1, RP1, and RP2 comparisons. Overall, 331 positions were hypermethylated (q-value < 0.01, *methylKit*); among these, seven demonstrated a statistically significant downregulation in methylation during RP1 (q-value < 0.01, *methylKit*). An additional two transcripts demonstrated a statistically significant downregulation during RP2 (q-value < 0.01, *methylKit*). 208 positions were noted to contain fewer m⁶A modifications (q-value < 0.01, *methylKit*). Amongst these, one position demonstrated increased methylation within RP1 (q-value < 0.01, *methylKit*). No additional positions demonstrated increased methylation within RP2 (q-value < 0.01, *methylKit*). Spaceflight had a measurable and sustained effect on m⁶A methylation profile of the astronauts for select genes.

7) For the new transcript that the authors identified- can they show the read count overlaid into the UCSC genome browser to enable the readers to get a sense of the transcript and its neighbourhood. Is the gene conserved in other organisms?

We have published and included a link to a UCSC Genome Browser Session for viewers interested in integrating the data for this particular gene for all samples with a wide variety genome-wide analyses. Based on this session, we determined that the transcript exists within an intronic region for one particular gene, which is exceedingly large and which contains several other genes approximately 10 kb upstream and downstream of the particular region of interest. The gene is unlikely to be conserved; however, conserved regions likely exist less than 1kb both upstream and downstream of the transcript alignments. We have additionally added Supplemental Figure S3, which displays the raw alignments informing this novel transcript variant within the Integrated Genomics Viewer and plainly demonstrates its raw base pair sequence. At the present, we have no speculation as to the functional implications of this finding, but these isoforms and RNAs will be the subject of future studies.

Moreover, the raw data is now online at NASA's Gene Lab repository, and we have made a preview link for the reviewers: <https://osdr.nasa.gov/bio/repo/data/studies/OSD-569/preview/SQ3QgEzNMH5q1X19jKWJxVLF5Lx5RQtp>.

We are very grateful to all reviewers for their insight, critique, new ideas, and help with our manuscript. Thank you.

REVIEWERS' COMMENTS

Reviewer #1 (Remarks to the Author):

The authors have satisfactorily responded to my inquiries. While their data interpretation remains largely descriptive, the dataset itself offers invaluable insights for the scientific community. And their findings are interesting. Furthermore, this implementation of ONT DRS highlights the importance of embracing emerging technologies in research. I strongly encourage the authors to promptly release their raw data. I currently have no additional queries.

-- Guanzheng Luo

Reviewer #2 (Remarks to the Author):

The authors have addressed my original comments thoroughly and well.

Minor comments:

Please define 'DRACH' or simply state that these are IUPAC ambiguity codes.

"These data indicate the *expression* levels of the genes responding to spaceflight are distinct from their specific RNA modifications, and that this is consistent across the immune and hematopoietic-related pathways." I think it would be more precise/less ambiguous here to state that the *transcription* levels are distinct, since methylation may affect translation which could be interpreted as a change in expression (not measured here).

Andrew Routh

Reviewer #3 (Remarks to the Author):

The authors have addressed my concerns. I don't have any questions for now.

Dear Reviewers and the editors,

We thank the reviewers for their careful review of our study and results, and we have updated our manuscript with updated figures, improved text, and additional validation data sets. We have responded to their questions below, with reviewer comments in black and our responses in blue.

Reviewer #1 (Remarks to the Author):

This pioneering study utilized ONT DRS to investigate the effects of spaceflight on human physiology at the epitranscriptome level. The authors analyzed DRS profiles and identified key genetic pathways that exhibited significant changes in four SpaceX Inspiration4 astronauts before spaceflight, upon return, and during recovery. Significantly, this is the first m6A methylation profile for a human space mission. The authors demonstrated that ONT could be a robust and efficient platform to monitor the molecular dynamics of astronauts during a flight mission. This study represents just the beginning of an accumulation of information that will lead to greater understanding of the human body and increased confidence in dealing with deep space travel in the future. Technically, the authors performed several routine analyses to identify changes in the transcriptome and epitranscriptome. However, I have some minor questions and suggestions for the authors:

We thank Reviewer #1 for recognizing the significance of this study and providing helpful feedback. We agree that this dataset provides the first-ever view of such data, and as such, will serve as a baseline data set for a deeper understanding of human physiological and transcriptome effects due to spaceflight. Please find attached our point-by-point responses and updates to the manuscript.

1. Can the authors provide a general comparison of their results with those of previous studies that used other sequencing platforms to investigate gene expression in astronauts?

Yes, we have now updated this section in the Introduction and Discussion, and we also provide a broader review of other studies. To comprehensively detail previous studies that investigated gene expression in astronauts, we reviewed the NASA GeneLab human transcriptomic studies and placed the following revised summary of previous studies in our introduction: “Ground studies collecting transcriptomic profiles of human subjects have attempted to characterize the impact of individual components of spaceflight, such as radiation or microgravity exposure and simulated with a parabolic flight. With regard to direct spaceflight exposure, the available transcriptomic data is even more limited, focused on a few specific assays, either sequencing the human IgM repertoire following long-term spaceflight, measuring the proteome of exhaled breath condensate after long-term spaceflight, or examining gene expression profiles from astronaut hair follicle samples after long stays on the International Space Station. However, beyond this limited work, there is scant data on the impact of human spaceflight on the transcriptome beyond the NASA Twins Study (1-year mission), and almost no data from short-duration missions. Moreover, there are no single-molecule, direct RNA datasets from

astronauts, which can help detail changes in RNA methylation and base modifications while also capturing expression and isoform changes.”

2. In Table 1, the authors found several unique transcripts at one time point. Is it possible that these genes were caused by low sequencing depth? Please clarify.

To mitigate any low-coverage issues with these data, we have created a minimum coverage threshold for each gene, as suggested by the authors of the *StringTie* program. On a per-sample basis, there exists some variability in coverage, with most samples mapping between 1 and 2 million LRS reads. These differences become less pronounced when comparing between time points. Therefore, in order to extract meaningful biological signals, we required that transcripts be shared among all samples for a given time point, where some samples had below-average coverage and some had above-average coverage, and that they not be shared by all other samples within the comparison set, spanning a similar diverse range of coverages. Given this analytical framework, we believe that the analysis is robust to differences in sequencing depth. We have additionally updated Supplemental Table S5 to provide metrics for per-sample read alignment and coverage. We now also include orthogonal validation data from a distinct NGS platform (Ultima UG100).

3. To illustrate the exemplified genes with changes, please provide an IGV snapshot or a visualizable gene structure and sequencing results in another form.

We have added Supplemental Figure S3, which displays the raw alignments informing this novel transcript variant and demonstrates its raw base pair sequence. Additionally, we have published and included a link to a UCSC Genome Browser Session for viewers interested in integrating the data for this particular gene for all samples with a wide variety genome-wide analyses.

4. The authors first used m6anet to identify the methylated sites, and then applied the methylKit tool, which was designed to analyze DNA methylation, to find the differentially methylated m6A profile. This appears unusual. I would suggest that they use another specialized tool to directly call diff-m6A. At the very least, they should compare results from different tools.

Since the submission of our article, a benchmarking study on m⁶A callers was published, “*Systematic comparison of tools used for m⁶A mapping from nanopore direct RNA sequencing*” (10.1038/s41467-023-37596-5), which demonstrates that m6anet is the most accurate m6a caller, and we have now cited in the Discussion: “While accounting for secondary mappings with *salmon* could be argued to be the more sound approach, this calls for further assessment and discussion of methods to be used for analyses of direct-RNA data, as retention of secondary mappings is incompatible with m⁶A callers such as *m6anet*, which was recently shown to be the most accurate m⁶A caller within a comparative software study.” Once m⁶A modifications have been called, there are even fewer standardized parameters for the downstream analysis of RNA modifications. We selected methylKit to call differences between the given probabilities, since it employs Fisher’s exact test, which is a suitable, non-parametric test for comparing groups with

small sample sizes. We lastly point to one prior publication which serves as precedent for employing this particular combination (10.1101/2022.01.24.477497).

In our estimation, m6anet was the best suited for our particular experimental design and sequencing data, since many other m⁶A callers, including nanocompore, differr, DRUMMER, JACUSA2, xPore, and yanocomp, require a negative control, low-m⁶A sample, or knock-out model. Of note, m6anet does not require a low-m6a profile because it is applying a machine learning model which is able to generally differentiate electrical signals corresponding to a modified base pair from an unmodified base pair. There are other similarly designed tools, including ELIGOS, EpiNano, nanom6A; however, they were trained on data from older chemistries and base-calling tools. We used m6anet since it does not have this limitation.

5. ONT has its own advantages over other sequencing platforms, especially in space. Can the authors expand their imagination and speculate on the future scenario of instant use of ONT in space?

We agree that this is a major advantage to the ONT platform and we have now edited our discussion to emphasize this (novel text in *italics*), with the first paragraph reading as follows: “All prior data from spaceflight studies have used short-read sequencing. The advent of direct-RNA sequencing in particular allows for additional intrinsic information about the RNA, such as m⁶A methylation, to be detected.

“The development of space-specific, microgravity-compatible sequencing protocols foreshadows the exciting possibility of real-time monitoring of changes of the transcriptome and epitranscriptome for future missions.”

We additionally discuss the experimental demonstration of direct-RNA sequencing, the same as in this paper, within the final paragraph of the discussion:

“Finally, given the recently proposed missions with private (SpaceX, Axiom, Sierra Space) and public (NASA, ESA, and JAXA) space entities, there will be continued opportunities to replicate these findings and to continue to discover new features of the transcriptome. This will include continued mapping of space-related gene expression responses, m6A site changes, and isoform-switching events that are associated with spaceflight. Also, in-flight testing of direct RNA sequencing has been demonstrated on the ISS, which indicates that LRS of RNA could also be applied for crew studies in future missions, including for blood, microbial, and environmental samples. Optimization and deployment of these single molecule technologies can also aid in the plans for lunar laboratories, in-flight clinical diagnostics, and the ability to discover non-canonical bases or new base modifications for lunar and exploration-class (e.g. Mars) missions.”

On a final, exciting note, we have just completed flight testing of the Mk1C device at SpaceX headquarters, which is being QC'd for upcoming missions and should hopefully fly to space within a year, and will enable us to test these protocols directly in flight.

Reviewer #2 (Remarks to the Author):

The investigators provide an transcriptomic analysis of samples collected from the I4 astronauts both before, immediately after and during recovery from their space flight. This is a unique and valuable resource that is here studied using a novel and interesting technology (long-read direct RNA sequencing on the ONT nanopore platform). Despite the obvious interest and impact of the study design, there are a few issues confounding a rigorous analysis and interpretation of the data, as described below that greatly limit the scope of this report. Together with a somewhat superficial scrutiny of the output data, this report appears to be somewhat preliminary.

We thank Reviewer #2 for their interest in this study and their informed comments. Throughout our analysis, we wished to present the results in a manner which did not overgeneralize, given limitations in terms of astronaut numbers. We aimed simply to provide a series of robust observations (especially changes observed in all crew members and with multiple pipelines), providing a useful roadmap for future transcriptomic analysis, as well as highlighting key genes and pathways, which can guide future spaceflight missions and experiments. In line with this general statement, we attach our point-by-point responses, and also highlight new data and analyses in our revision.

Major Comments:

A pervading issue in this report is in how individual datasets are treated during statistical analyses. It is not clearly stated, but I assume each individual is treated as a 'biological' replicate, thus yielding $n=4$ for most samples. If technical replicates were obtained in this study, it is not clearly described. Some clarity on whether this process would satisfy the assumptions made by the underlying statistical tests is warranted. Related to this, given the small size of the cohort (4 individuals) some discussion of how inter-person variability (e.g. age, genetic background, biological sex) might impact these studies since there is also likely to be considerable intra-subject variability.

Indeed we used 4 biological replicates for each time point, and then compared between groups. We have clarified this within the manuscript by adding the following information to the presentation of **Supplemental Table S1**: "Both pipelines were applied to the data in the context of multiple flight, recovery, and longitudinal profiles (**Supplemental Table S1**). A single sequencing run was performed for each astronaut and time-point; the four astronauts were grouped together as biological replicates for downstream analysis." Ultimately, technical replicates are not an input recommendation or requirement for *edgeR*, *DESeq2* or *methyKit*.

As we have processed the data, in accordance with best practices, we have only utilized de-identified data. Therefore, we cannot specifically associate observed inter-person variability with the personal health information of the astronauts themselves. Of note, the flight subjects were considered as covariates in the differential expression analyses and linear models, to account for inter-person variability. We have clarified this within the Methods section of the manuscript: "Covariates to account for inter-astronaut variability were included when performing both *DESeq2* and *edgeR* analysis."

Also, we have added a section in the Discussion about this important point: “Moreover, given the civilian background of the I4 astronauts, we note that transcriptomic changes can be confounded with age, biological sex, preflight preparation differences, or divergent postflight recovery procedures, and thus utilized tools that aim to model and normalize such variation.”

Similarly, since the RNA studied is derived from whole-blood collections, changes in the astronaut blood chemistry/cell counts/etc is likely to be very different between individuals and may skew the abundance of genes/gene sets found for each individual. For example, should one individual exhibit mild leukocytosis after the space flight, then more RNA transcripts/gene sets associated with white blood cells would be found, rather than any actual changes in the expression profiles of these genes. As a result, without supporting information and/or characterization of the samples collected and tested, the investigators cannot distinguish between changes in transcriptomic/epitranscriptomic profiles due to gene expression regulation, versus capture of different transcriptomic/epitranscriptomic features due to changes in the complexity/compositions of the samples being studied.

This is an important consideration, and we now include here results included in a companion paper (Kim et al) that used FACS to characterize cell proportions from the mission. We also now include the complete blood count (CBC) and comprehensive metabolic panel (CMP) results from the CLIA lab (Quest) blood processing that was performed on the crew, which detailed the proportion of cell types found in the blood at all time points. Both methods showed no significant difference (Wilcoxon Rank Sum, p-values all >0.1) in the proportion of the cell types in the blood in pre- and post-flight, which should minimize the impact of cell type representation in the blood transcriptome data.

Reviewer Figure 1. PBMC subpopulation markers for i4 crew samples confirmed with FACS. (left) FACS gating of T cell, B cell, NK cell, and monocytes from the i4 PBMCs and (right) Cell proportion of i4 PBMC over time calculated from FACS, with non-significant (NS) results for the proportions of cell types before and after spaceflight.

a**Complete Blood Count****b****Comprehensive Metabolic Panel**
Reviewer Figure 2. CBC and CMP profiles of the i4 crew before and after spaceflight. Each dot represents each crew member (different colored dots for C001, red, C002, green, C003, blue, C004, purple, listed on the legend on the right side) for the Complete Blood Count (CBC) profile (top) and Comprehensive Metabolic Panel (CMP, bottom) panel, with cell types and analytes annotated on top of each plot. Measured analytes were all within nominal range (white zones) or within 5-10% of the nominal range.

This study is limited by the lack of any in vitro validation of the observed genes in gene expression and/or isoform abundance. While RT-PCR validation of differentially expressed genes is not necessarily required since this is a pretty well established technology; the small number of novel isoforms, including novel isoforms with putative retain introns (for example) warrants follow-up studies (if there are biological materials still remaining). This point is particularly important given the short length of many of the 'long-reads' collected in this study and that the unique technology used in this study is long-read sequencing. Incomplete capture of full-length transcripts in the direct RNA sequencing protocol may well confound the interpretation and robustness of the downstream computational analysis (as eluded to in the discussion).

We have verified the existence of *de novo* transcript variants within the Ultima Genomics deep-RNA sequencing platform, verifying expression within the embedded genomic loci. We verified the expression of 23 out of 25 (92%) transcripts utilizing this technology, which provides a strong basis for their validity and is, by itself, an in-depth RNA-seq data set for future analysis and studies.

We agree that future samples characterization will be important to understanding these RNA changes' physiological relevance for spaceflight. Given the nature of *de novo* transcriptome analysis, we detected specific reads which were unique to a specific time-point across astronauts; however, we made no speculation as to the physiological relevance of these findings. Therefore, among the 25 different transcripts identified in **Table 1**, we can only use predicted functions. We plan to biobank all remaining biological material, to ensure its availability for future comparative studies. Indeed, we hope to replicate similar trends via *de novo* transcriptome and deep RNA sequencing analysis for the Polaris Dawn mission (<https://polarisprogram.com/dawn/>), and thereafter return to genes of interest appearing across two separate astronaut missions into low-earth orbit.

Minor Comments:

The investigators do not provide a concise description of the samples collected or study design. While this information can be gleaned from the supplementary tables and may well be described elsewhere in other associated manuscripts, some text describing the basic cohort, timing of collections, type of samples collections, number of replicates, etc, etc would be helpful, even if this information is provided elsewhere. For example 'R+1' is not defined (although it is quite obvious what this would be, it would be good to provide this information directly). Similarly, how the current study is distinct from the Overbey manuscript (i.e. different samples/different technological focus) would be important, since it appears that this is a companion paper.

Thank you for this comment. We have now posted a complete and in-depth description of the protocols used in this paper, as well as the connection to other assays and samples, in a separate submission and companion paper, "Collection of Biospecimens from the Inspiration4 Mission Establishes the Standards for the Space Omics and Medical Atlas (SOMA)."

<https://www.ncbi.nlm.nih.gov/pmc/articles/PMC10187258/>. We have clarified this connection at the top of the **Methods section**: “Methods // **Direct RNA sequencing**. Total RNA was processed and sequenced as described in Collection of Biospecimens from the Inspiration4 Mission Establishes the Standards for the Space Omics and Medical Atlas (SOMA).”

Moreover, we have updated the terminology (e.g. R+1) and clarified the text. Specifically, we have included the following sentence in the **Introduction**: “We here analyze LRS samples taken before (L-92, L-44, L-3 days) and after (R+1, R+45, R+82, R+194 days) spaceflight.” As previously mentioned, we have clarified the treatment of replicates by adding the following information to the presentation of **Supplemental Table S1**: “Both pipelines were applied to the data in the context of multiple flight, recovery, and longitudinal profiles (**Supplemental Table S1**). A single sequencing run was performed for each astronaut and time-point; the four astronauts were grouped together as biological replicates for downstream analysis.”

Also related to this, since this is a ONT-focused manuscript, some descriptions of the datasets collected would be appropriate before launching into the results of the data analysis. E.g. number of reads, number of mapped reads, length of reads, etc, etc. Again, although this information might be gleaned from supplementary materials, some descriptive statements on the data collected and data quality and how these compare to other publicly accessible datasets or standards would be helpful.

These are also important points, and we have updated the manuscript to help provide clarity. We have also added more details to the text and the supplemental tables, to help clarify these points. Specifically, **Supplemental Table S5** contains information regarding the number of mapped reads, transcript sensitivity and coverage. Based on this, we have specifically included the following sentence at the beginning of the results section: “All runs exhibited a minimum of 10-times transcriptome coverage post-alignment.” We include information regarding gene lengths within the **Methods** section: “Additionally, the sub-2Kbp read lengths (median length between 517 and 1,331, with one outlier of 279, Supplemental File S1) were insufficient to perform differential transcript usage analyses, and we focused on the differential gene expression results, which pipeline-transcriptome-de generates with edgeR while collapsing transcripts into respective genes.”

The relationship between the number of total reads and mapped reads is based on both stochastic chemical sequencing processes and initial RNA concentration input; we have verified this by replacing the reference genome with the Nanopore RNA calibration strand FASTA file, showing that nearly all unmapped reads map to this sequence, which is included by default within the Nanopore Direct RNA Sequencing kit, SQK-RNA002 (<https://help.nanoporetech.com/en/articles/6632031-what-is-rna-cs-racs>).

Given the novelty of the technique, there are few publications related to particular direct-RNA standards. More than 80% of publicly available datasets do not include deposition of raw FAST5 files, which limits the potential to perform a meta-analysis of data quality across publicly available datasets. There are approximately 500 samples deposited within the European

Nucleotide Archive (ENA) which include fast5 files; however, we believe that a meta-analysis to check for data quality is outside of the scope of this study. Since we have implemented the most recently available sequencing chemistry and basecalling methodology, we are confident that our technical data quality matches current best-practices. The basecalling metrics included in Supplemental File S1 suggest excellent Nanopore base-calling quality, according to manufacturer's recommendations and ENA best practices.

Some discussion/comments on why there are numerous genes that show no DE with the SARTools pipelines but do show changes with the ONT pipeline would be valuable (i.e. the red points along the x=0 axis). Is this due to the multiple mapping allowed by the ONT pipeline?

We have commented now that such differences are likely due to the heuristics in the algorithms, rather than multi-mapping reads, and this is clarified in the text. Specifically, we have addressed this in the Discussion: "Additionally, the associated differential expression callers, *DESeq2* and *edgeR*, respectively, utilize slightly different heuristics to perform \log_2 fold change estimates, resulting in many genes uniquely identified by the *salmon-edgeR* pipeline mapping to the y-axis of Figure 1b."

The vertical text in Figure 2B is hard to read and some definitions are missing. For example, "NEBEN_AML_WITH_FLT3_OR_NRAS_UP" is ill-defined. While I appreciate there are many pathways to be considered, defining the top few (as depicted in Figure 2B) with intuitive labels would be useful.

Thank you - we have updated the Figure for readability. Of note, the pathway names we used are exactly as they appear in the MSigDb database, to enable easier comparison with other datasets. In order to ensure the reproducibility of the figure, we need to match the name with the original database identifier. Nevertheless, we have added the following sentence to each figure legend referencing these labels: "Gene sets can be explored further at: <https://www.gsea-msigdb.org/gsea/msigdb>." In this manner, researchers can navigate to the Broad Institute page and utilize the search feature to better understand the internal features of these gene sets. For the case of "NEBEN_AML_WITH_FLT3_OR_NRAS_UP", the relevant page would be: https://www.gsea-msigdb.org/gsea/msigdb/human/geneset/NEBEN_AML_WITH_FLT3_OR_NRAS_UP.html?keywords=NEBEN_AML_WITH_FLT3_OR_NRAS_UP

In figure 2A and 2B, without any labels for what each row corresponds to, the images are impossible to interpret or scrutinize, other than to provide a broad overview of some (correlated?) changes in the expression levels or some genes amongst the cohort.

We have updated these figures. Specifically, we have added gene names to the heatmap, providing labels for each row.

For the section describing the m6A analysis, some high-level description of the number of putative m6A sites identified in each dataset and how many genes/transcripts show putative

changes, and how many of these correlate with the DE analysis performed in the previous section is needed. Some of this information is found in Figure 4, but could be expounded upon.

We have incorporated these statistics into the manuscript in the following paragraph: “Next, we identified the sites of m⁶A modification across isoforms and their degree of change, leveraging the single molecular nature of the ONT platform to find sites differentially methylated for the flight, recovery, and longitudinal profiles (q-value < 0.01, *methylKit*³⁰) (Supplemental Table S4). Herein, we recorded 1,190 m⁶A modification sites that demonstrated a statistically significant difference for at least one particular longitudinal comparison. These sites were located on 200 total transcripts, sourced from 193 gene loci.”

Comparing the 193 gene loci which contained DRACH motifs with differentially methylated sites (q-value < 0.01, *methylKit*) with the set of 440 differentially expressed gene loci (FDR-adjusted p-value < 0.05, *either salmon or SARTools*), we find a modest overlap of 53 genes. The relationship between gene expression and m⁶a methylation is still being examined, although it is likely a key regulator of mRNA decay. Within our analysis, we postulated on the functional implications of these shared genes by comparing gene sets, as opposed to individual genes.

When comparing R+1 to all pre-flight timepoints, is each timepoint from each individual treated as though they are independent replicates? This should be clearly stated in the maintext, in addition to the methods.

This has been updated in the methods; thank you for the careful reading. Specifically, we have added the following sentence to the **Differential Gene Expression Analysis** methods section: “For comparisons where multiple time-points were grouped as either preflight or postflight, each sequence run was treated as an independent replicate.”

In figures 3 and 4, much of the small text is difficult to read and should be reworked for clarity.

We have updated these figures to improve clarity. Thank you.

In the discussion, the authors state "LRS can precisely identify such differences, given that the nearly all 257 transcripts are shorter than the theoretical maximum sequencing length for Nanopore platforms, which extends to nearly 2.3Mb". This statement is somewhat mis-leading, since the major limitations to capturing full-length RNA (or cDNA) transcripts are factors such as RNA fragmentation/integrity, success of long-range reverse-transcription (often limited to 4-5kb by standard RTs) and the quality of the final library loaded onto a flowcell.

We thank the reviewer for highlighting this statement; we agree that it requires additional context. We have altered the relevant section to read as follows: “LRS can potentially identify all such differences, given that nearly all transcripts are shorter than the theoretical maximum sequencing length for Nanopore platforms (2.3Mb). Nevertheless, limitations remain within the library preparation and technical specifications (e.g. 3' enrichment) of different LRS platforms,

and improvements in these protocols and informatics methods will help profiling for future samples.”

In terms of the specific issues mentioned above, we believe that these are generally applicable to all kinds of sequencing approaches, including second generation short-read sequencing and third-generation long-read sequencing. In the context of our study, long-range reverse-transcription is an optional component of the sequencing protocol, and therefore not a major concern in terms of transcript length. We believe that we have identified and expanded upon the major technical bias confounding transcript-level analysis within the discussion paragraph regarding 5' end truncation. Reference 44, which cites an article entitled *Opportunities and challenges in long-read sequencing data analysis* (10.1186/s13059-020-1935-5) provides in our estimation an excellent review of technical bias as they apply for all commercially available long-read sequencing platforms.

Reviewer #3 (Remarks to the Author):

The manuscript from Grigorev K. et al describes the transcriptome and m6A methylome of 4 astronauts on land, in space, and after returning. While this is an invaluable dataset that has incredible potential to be interesting, unfortunately, the current way that the manuscript is presented makes it incredibly boring. I appreciate that the authors were rigorous in using different methods to analyze the data, however, they did not seem to be able to extract much meaningful biology out of this dataset.

We thank Reviewer #3 for recognizing the significant interest in this dataset. Our first draft was perhaps too cautious in our interpretation; nevertheless, we hope that the mentioned rigorous analytical approach will provide grounds for robust biological interpretation, especially when this data is placed in context of future similar spaceflight observational studies, and in conjunction with the proliferation of commercial spaceflight efforts. Moreover, we have updated the discussion to highlight the significance of the study and the importance of the R+1 observations at the gene and first-ever observation of m⁶A level dynamics in astronauts.

Here are my major comments:

1) What is the distribution of the biotype of the genes that are disrupted upon flight- are they mostly coding, or are there also noncoding RNAs? How many transcripts did the authors identify?

We have clarified this in the text, and below, to note that almost all genes were protein-coding. Within the time-points we believed fit the description of demonstrating “disruption upon flight” - FP1, RP1, RP2, RP4, RP5 and RP6, further defined in Supplemental Table S1 - we noted that there were 130 transcripts that were differentially expressed. Based on the technical specifications of the Nanopore Direct RNA-sequencing protocol (SQK-RNA002), only poly-A tailed RNAs are robustly captured by the assay. We therefore observe differential expression of primarily protein-coding genes (green) as shown in the figure below.

Reviewer Figure 3. Gene annotation of types of genes showing expression and methylation changes by timepoint. Each gene (y-axis) is annotated by the time point of differential change (column) and gene type (gene= protein-coding, orange= rRNA, brown=mitochondrial, blue=lncRNA, pink=pseudogene, light-green=snRNA, yellow=scaRNA).

2) What percentage of the transcripts that showed expression changes during flight were recovered postflight? Are there transcripts that recovered faster versus slower?

We observed that approximately 90% of transcripts that demonstrated expression changes were recovered postflight. Given the short duration of the actual low-earth orbital spaceflight, we saw that all transcripts were recovered after 45 days, which was the first sampling time post spaceflight. Therefore, we did not have the resolution to examine the actual time velocity vector for transcript recovery. Ultimately, we analyzed the functional implications of these gene changes within the context of pathway analysis, as further described in Figure 3.

3) Are there shared regulators (such as RBPs) for the transcripts that recover or show similar expression changes?'

We thank the reviewer for this suggestion. We chose to utilize ChEA3 to answer this question (<https://maayanlab.cloud/chea3/>), since it integrates information from fairly recent CHIP-seq, RNA-seq and gene set databases to generate a holistic score. Within the manuscript, we have added the following section into the Results section: "Transcription factor enrichment analysis was then utilized to identify commonly shared transcription factors by genes (Supplemental Table S2) across all time points. The top three suggested regulators, based on a combined CHIP-seq and co-expression analysis, KLF1, GATA1, and TAL1 are all key transcription factors (TFs) for erythrocyte differentiation^{27–29}, supporting the changes in these cell types and also suggesting possible TF-based drivers for the trends observed within the GESECA analysis (Supplemental Table S3)." Likewise, we have added the following section to our Methods section: "**Transcription factor enrichment analysis.** Gene annotations within **Supplemental Table S2** were converted to gene names utilizing gProfiler (<https://biit.cs.ut.ee/gprofiler/convert>).⁵¹ The resulting gene list was input into the ChEA3 digital web server (<https://maayanlab.cloud/chea3/>).⁵² The mean rank results were exported and reported in **Supplemental Table S3**." In our experience, programs which predict regulation are heavily reliant on their particular source of data. For example, programs such as BART (<https://zanglab.github.io/bart/>) and ChIP-Atlas (https://chip-atlas.org/enrichment_analysis), provide regulatory profiles highly dependent on the contexts of the sourced CHIP-seq data, which may not be applicable to spaceflight contexts, which is another reason we used ChEA3.

4) For the m6A differential sites- what is the distribution of their changes along a transcript?

For readers to better examine questions related to the differential m6A profiles, we have updated **Supplemental Table S4** to include the definitions for the thirteen different comparisons contained within, noting redundancy with **Supplemental Table S1**. *m6anet*, the m6a modification quantifier for Nanopore Direct RNA-sequencing data, was utilized to obtain per-site methylation probabilities. *m6anet* only makes predictions on transcript positions with DRACH motifs, which encompass the predominant consensus motifs for m6a methylation in humans. Each DRACH motif needs at least 20 reads of coverage in order to be eligible for a prediction. Specific predicted DRACH motif positions were included in **Supplemental Table S4** if their *methylKit* q-value fell below 0.01 for at least one comparison.

Based on these criteria, we identified 1,190 DRACH motifs that demonstrated a statistically significant difference in one particular comparison time-point. These DRACH motifs were located on 200 total transcripts, sourced from 193 gene loci. However, since our long-read data exhibited 5' truncation, due to polyA-tail priming, we ultimately chose to consider these m⁶A differences as discrete units; either in terms of transcript location or transcript identity. Unfortunately, assessing for differences within different gene segments, such as the 5' UTR, CDS and 3' UTR for protein-coding genes, will likely reveal changes caused by technical truncation, as a function of the transcript length, as opposed to true biological differences.

We have incorporated these specific statistics into the manuscript in the following paragraph: "Next, we identified the sites of m⁶A modification across isoforms and their degree of change, leveraging the single molecular nature of the ONT platform to find sites differentially methylated for the flight, recovery, and longitudinal profiles (q-value < 0.01, *methylKit*³⁰) (**Supplemental Table S4**; OSDR accession number OSD-569). Herein, we recorded 1,190 m⁶A modification sites that demonstrated a statistically significant difference for at least one particular longitudinal comparison. These sites were located on 200 total transcripts, sourced from 193 gene loci."

5) For the genes with differential m⁶A modifications- do they show gene expression changes?

Comparing the 193 gene loci which contained DRACH motifs with differentially methylated sites (q-value < 0.01, *methylKit*) with the set of 440 differentially expressed gene loci (FDR-adjusted p-value < 0.05, *either salmon or SARTools*), we find a modest overlap of 53 genes. The relationship between gene expression and m⁶A methylation is still being examined, although it is likely a key regulator of mRNA decay. Within our analysis, we postulated on the functional implications of these shared genes by comparing gene sets, as opposed to individual genes. This has also been updated in the manuscript.

6) Are the m⁶A modifications recovered post-flight?

This is a great question, which we have now added to our manuscript. To answer this question, we examined the m⁶A differences observed within the FP1, RP1, and RP2 comparisons. Overall, 331 positions were hypermethylated (q-value < 0.01, *methylKit*); among these, seven demonstrated a statistically significant downregulation in methylation during RP1 (q-value < 0.01, *methylKit*). An additional two transcripts demonstrated a statistically significant downregulation during RP2 (q-value < 0.01, *methylKit*). 208 positions were noted to contain fewer m⁶A modifications (q-value < 0.01, *methylKit*). Amongst these, one position demonstrated increased methylation within RP1 (q-value < 0.01, *methylKit*). No additional positions demonstrated increased methylation within RP2 (q-value < 0.01, *methylKit*). Spaceflight had a measurable and sustained effect on m⁶A methylation profile of the astronauts for select genes.

7) For the new transcript that the authors identified- can they show the read count overlaid into the UCSC genome browser to enable the readers to get a sense of the transcript and its neighbourhood. Is the gene conserved in other organisms?

We have published and included a link to a UCSC Genome Browser Session for viewers interested in integrating the data for this particular gene for all samples with a wide variety genome-wide analyses. Based on this session, we determined that the transcript exists within an intronic region for one particular gene, which is exceedingly large and which contains several other genes approximately 10 kb upstream and downstream of the particular region of interest. The gene is unlikely to be conserved; however, conserved regions likely exist less than 1kb both upstream and downstream of the transcript alignments. We have additionally added Supplemental Figure S3, which displays the raw alignments informing this novel transcript variant within the Integrated Genomics Viewer and plainly demonstrates its raw base pair sequence. At the present, we have no speculation as to the functional implications of this finding, but these isoforms and RNAs will be the subject of future studies.

Moreover, the raw data is now online at NASA's Gene Lab repository, and we have made a preview link for the reviewers: <https://osdr.nasa.gov/bio/repo/data/studies/OSD-569/preview/SQ3QgEzNMH5q1X19jKWJxVLF5Lx5RQtp>.

We are very grateful to all reviewers for their insight, critique, new ideas, and help with our manuscript. Thank you.

Dear Editors,

Thank you for sharing the final reviewer comments. We have responded to their comments below, with reviewer comments in black and our responses in blue.

Reviewer #1 (Remarks to the Author):

The authors have satisfactorily responded to my inquiries. While their data interpretation remains largely descriptive, the dataset itself offers invaluable insights for the scientific community. And their findings are interesting. Furthermore, this implementation of ONT DRS highlights the importance of embracing emerging technologies in research. I strongly encourage the authors to promptly release their raw data. I currently have no additional queries.

-- Guanzheng Luo

We thank Dr. Luo for his insightful review of our article. The raw data for this study will be published at the time of publication within the NASA GeneLab repository at the following address: <https://osdr.nasa.gov/bio/repo/data/studies/OSD-569>

Reviewer #2 (Remarks to the Author):

The authors have addressed my original comments thoroughly and well.

Minor comments:

Please define 'DRACH' or simply state that these are IUPAC ambiguity codes.

"These data indicate the *expression* levels of the genes responding to spaceflight are distinct from their specific RNA modifications, and that this is consistent across the immune and hematopoietic-related pathways." I think it would be more precise/less ambiguous here to state that the *transcription* levels are distinct, since methylation may affect translation which could be interpreted as a change in expression (not measured here).

Andrew Routh

We thank Dr. Routh for his detailed review of our article. We have added a sentence defining the DRACH sequence motif at the time of invocation: "Comparing the 193 gene loci which contained DRACH (D=A, G or U; H=A, C or U) sequence motifs³³ with differentially methylated sites"

33. Meyer, K. D. *et al.* Comprehensive Analysis of mRNA Methylation Reveals Enrichment in 3' UTRs and near Stop Codons. *Cell* **149**, 1635–1646 (2012).

We agree with Dr. Routh's suggestion in specifying transcription as opposed to expression, since we did not integrate these data with proteomic assays.

Reviewer #3 (Remarks to the Author):

The authors have addressed my concerns. I don't have any questions for now.

We thank the reviewer for his comments and inquiries.